https://doi.org/10.1038/s41467-021-23721-9　　**OPEN**

# Layered double hydroxide membrane with high hydroxide conductivity and ion selectivity for energy storage device

Jing Hu[1,2,4], Xiaomin Tang[2,3,4], Qing Dai[1,2], Zhiqiang Liu [3], Huamin Zhang[1], Anmin Zheng [3✉], Zhizhang Yuan[1✉] & Xianfeng Li [1✉]

Membranes with fast and selective ions transport are highly demanded for energy storage devices. Layered double hydroxides (LDHs), bearing uniform interlayer galleries and abundant hydroxyl groups covalently bonded within two-dimensional (2D) host layers, make them superb candidates for high-performance membranes. However, related research on LDHs for ions separation is quite rare, especially the deep-going study on ions transport behavior in LDHs. Here, we report a LDHs-based composite membrane with fast and selective ions transport for flow battery application. The hydroxide ions transport through LDHs via vehicular (standard diffusion) & Grotthuss (proton hopping) mechanisms is uncovered. The LDHs-based membrane enables an alkaline zinc-based flow battery to operate at 200 mA cm$^{-2}$, along with an energy efficiency of 82.36% for 400 cycles. This study offers an in-depth understanding of ions transport in LDHs and further inspires their applications in other energy-related devices.

[1] Division of Energy Storage, Dalian National Laboratory for Clean Energy, Dalian Institute of Chemical Physics, Chinese Academy of Sciences, Dalian 116023, China. [2] University of Chinese Academy of Sciences, Beijing 100049, China. [3] State Key Laboratory of Magnetic Resonance and Atomic and Molecular Physics, National Center for Magnetic Resonance in Wuhan, Key Laboratory of Magnetic Resonance in Biological Systems, Wuhan Institute of Physics and Mathematics, Innovation Academy for Precision Measurement Science and Technology, Chinese Academy of Sciences, Wuhan 430071, China. [4] These authors contributed equally: Jing Hu, Xiaomin Tang. ✉email: zhenganm@wipm.ac.cn; yuanzhizhang@dicp.ac.cn; lixianfeng@dicp.ac.cn

I n addition to conventional membrane separation processes[1,2], there is a dramatically increasing demand for ion transport membranes in energy storage field, which is the key technology to address the issues of intermittency and instability of renewable energies like wind and solar power[3–5]. The flow batteries are well suitable for large-scale energy storage with their best combination of security, efficiency, and flexibility. As a key component of flow battery, the high-performance membranes have spurred considerable research, which play the vital role of impeding active materials, while conducting charge-balancing ions to form the internal circuit[6–8]. Among them, porous membranes are proved to be a promising choice[9–11]. Except for the general advantages of high chemical and physical stability[12–14], ideal porous membranes should have both high ions selectivity and conductivity for industrial application.

At present, studies on size-based separation membranes are mainly focused on zeolites[15,16], metal-organic frameworks (MOFs)[17–19], covalent organic frameworks (COFs)[20,21], graphene-based materials[22–24], ionic clays[25], and microporous polymers[10,26–28]. Among them, LDHs, a representative of 2D anionic clays, have attracted extensive interest in catalysis, bioengineering, and exhibited promising prospects for separation process due to their well-defined interlayer galleries, which can realize precise molecular/ion sieving[29]. Unlike stacked graphene or graphene oxide sheets, the interlayer spacing for LDHs can be adjusted by the replacement of anions.

In contrast to the extensive studies of LDHs for gas separation, ion separation and the transport behavior of ions through LDHs and further their application in energy storage have been rarely investigated. Recently, the proton transport behaviors through 2D-layered materials such as graphene oxide (GO)[30–32] and hexagonal boron nitride (h-BN)[33] have been uncovered, manifesting that the proton transport could be realized along 2D surfaces or interlayer channels. It is worth noting that LDHs belong to the relatively rare category of hydroxide ion conductors benefiting from the abundant hydroxyl groups bound by covalent bonds in the 2D host layer. Combing with their well screened channel that can selectively separate different ions with different size and strong hydrogen bond network for hydroxide ions transport, LDHs are believed to ensure the membrane with proper ion selectivity as well as fast ions transport in alkaline environment, which is expected to break the trade-off between selectivity and permeation for a membrane and further enable a high-performance separation process. Although several groups have demonstrated high hydroxide ion ($OH^-$) conductivity of LDHs and explored the prospect of using LDHs in alkaline environment[34,35], the deep-going studies of LDHs in the field of ions transport and the transport mechanism of $OH^-$ along ordered hydrogen-bonded nanostructures are very limited, which hindered the application of LDHs in ion separation field.

Here, we show a MgAl-based LDHs composite membrane with fast and selective ions transport for flow battery application. Combining the well-defined interlayer gallery with a strong hydrogen bond network along 2D surfaces, a high selectivity and superb hydroxide ion conductivity in an order of $10^{-2}\,S\,cm^{-1}$ can be achieved. Ab initio molecular dynamics (AIMD) simulations provide direct information about the transport behavior of $OH^-$ in restricted interlayer gallery of LDHs, revealing that the fast hydroxide ions transport behavior in LDHs channels is attributed to the mutual effect between the hydroxyl groups, interlayer anions, and water molecules in the gallery (Fig. 1). As a platform for verifying the practicability of LDHs-based membrane, an alkaline zinc-iron flow battery (AZIFB) assembled with the designed membrane was investigated in detail, demonstrating a high coulombic efficiency (CE) of over 98% and an energy efficiency (EE) of over 82% at a current density of 200 mA cm$^{-2}$,

demonstrating a very competitive performance among recently reported zinc-based flow batteries. We demonstrate the selective ion transport of LDHs composite membrane and their application in efficient and stable alkaline-based flow batteries, this study may inspire the utility of LDHs in other energy-related devices and enrich the development of membranes.

## Results

**Synthesis of LDHs and LDHs-based membrane.** LDHs has the general formula of $[M_{1-x}^{2+}M_x^{3+}(OH)_2]^{x+}(A^{n-})_{x/n} \cdot mH_2O$ ($M^{2+}$, $M^{3+}$, $A^{n-}$, and $H_2O$ represent di- and tri-valent metal ions, $n$-valent anions and the interlayer water, respectively). As shown in Fig. 2a, exemplified with $Mg_2Al(OH)_6Cl \cdot 2H_2O$ (referred as MgAl-Cl-LDH), MgAl-Cl-LDH consists of positively charged brucite-like host layers and interlayer galleries containing charge compensating anions. Metal cations ($Mg^{2+}$, $Al^{3+}$) and hydroxide ions are located in the centers and vertexes of octahedra, respectively. Homogeneous MgAl-Cl-LDH nanoparticles were synthesized using the co-precipitation and hydrothermal treatment method (see details for synthesis of LDHs in methods)[36]. As clearly indicated by environmental transmission electron microscopy (ETEM) image in Fig. 2b and Supplementary Fig. 1, the neighboring host layers of prepared MgAl-Cl-LDH presented an interplanar spacing of 0.768 nm for (003). Anion layers can be directly observed using the ETEM technology (Fig. 2b), presenting an interplanar spacing of 0.389 nm for (006) in high-resolution image. The prepared MgAl-Cl-LDH nanoparticles exhibit plate-like structure and almost hexagonal shape (Fig. 2c). And the lateral dimension of MgAl-Cl-LDH nanoparticles was found in the range of 100–200 nm, which was in good agreement with the dynamic light scattering (DLS) data (Supplementary Fig. 2).

To obtain a LDHs-based composite membrane, a poly(ether sulfone) (PES) porous membrane with a thickness of about $105 \pm 5\,\mu m$ and a porosity of 61.4% was employed as the support (referred as Substrate). The morphology of the support was adjusted by using hydrophilic sulfonated poly(ether ether ketone) (SPEEK) during phase inversion procedure. A smooth surface and an asymmetrically finger-like cross-section morphology could be found for the support (Supplementary Fig. 3). The PES substrate with asymmetric morphology and nanoscale pores generally possesses high conductivity but low selectivity, which is suitable for being the support of the composite membrane (Supplementary Fig. 4a). After spraying-coating the synthesized MgAl-Cl-LDH nanoparticles onto the support, the LDH nanoparticles that consist of magnesium, aluminum, chlorine, and oxygen atoms equably overlay on the surface of the membrane in a well-aligned manner (Fig. 2d and Supplementary Fig. 5), forming a LDHs-based composite membrane (abbreviated as LDH-M) with nanopores, which shows a higher specific surface area than substrate (Supplementary Fig. 4b, c). The lamellar LDHs layer shows a thickness of about $15 \pm 1\,\mu m$ (Fig. 2e). In short, the uniformly arranged LDHs flake layer with a well-defined interlayer gallery is expected to sieve target ions through the membrane effectively and a hydrogen bond network among the hydroxyl groups, interlayer anions ($OH^-$), and water molecules in the gallery of LDHs can guarantee the fast hydroxide ions transport.

**Ion selectivity and conductivity.** Highly selective membranes with well-tuned channel size are crucial for separation process, since the separation efficiency based on size exclusion for 2D layered material is ascertained by the relation of d-spacing to the size of the target species in a given system. For the designed composite membrane, the 2D layered LDHs were arranged in an

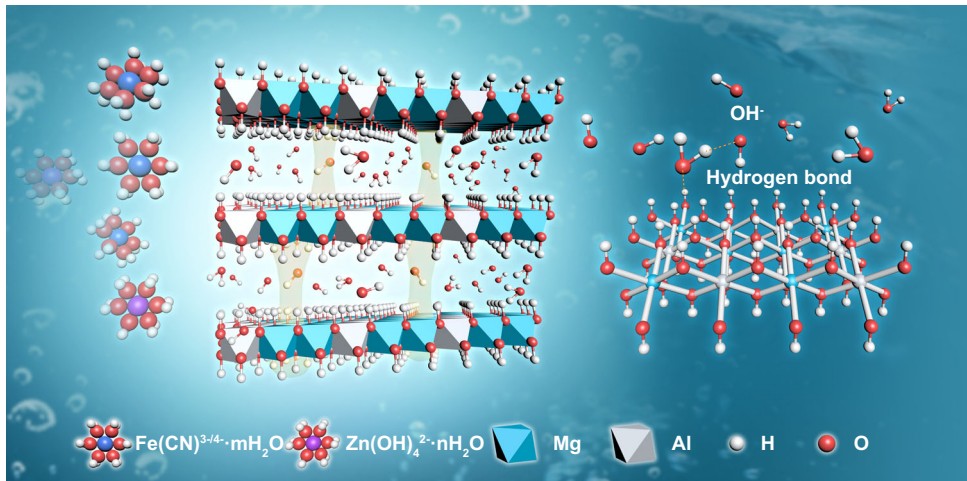

**Fig. 1 Selective ions transport and the hydroxide ions transport mechanism for LDHs.** The well-defined interlayer spacing of LDHs plays an important role in selective ions transport. The strong hydrogen bond network in the gallery of LDHs is expected to facilitate the effective conduction of hydroxide ions.

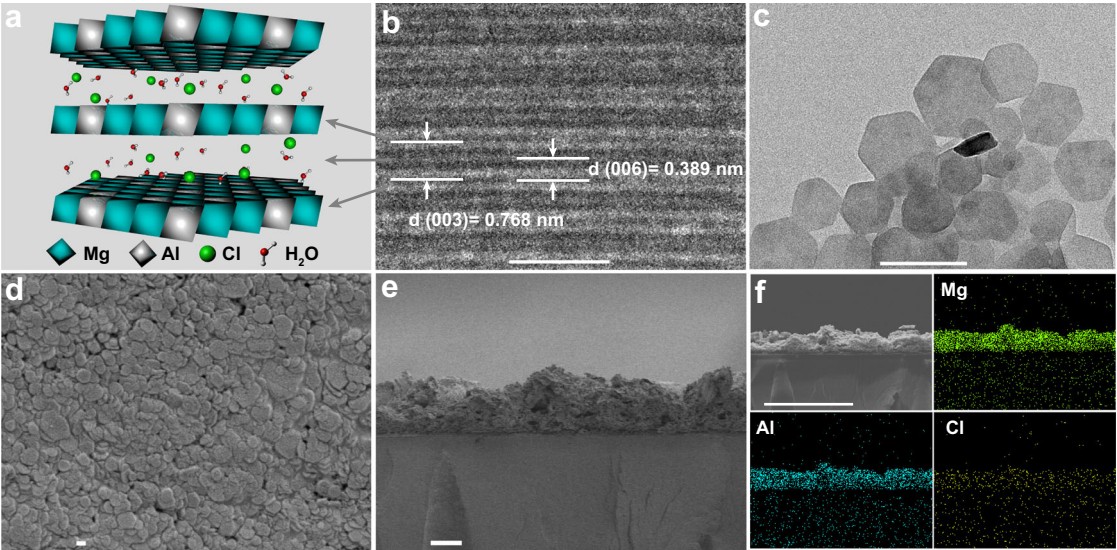

**Fig. 2 Layered double hydroxide composite membranes. a** Schematic structure of MgAl-Cl-LDH. **b** Environmental transmission electron microscopy (ETEM) image and **c** TEM image of MgAl-Cl-LDH nanosheets. **d** The surface morphology and **e** cross-section morphology of LDHs-based membrane. **f** The corresponding energy dispersion spectroscopy (EDS) analysis of its elemental distribution on LDHs flake layer. Scale bars for **b**, **c**, **d**, **e**, and **f** are 2 nm, 200 nm, 100 nm, 10 μm, and 60 μm, respectively.

ordered manner on the porous support (Figs. 2d, e and 3a). LDHs layer in the membrane can achieve preliminary screening and extend the diffusion path of ions across the membrane thus improving the ion selectivity of the composite membrane. Furthermore, the strong hydrogen bond network in the gallery is expected to facilitate the effective conduction of hydroxide ions, thereby ensuring the composite membrane with an excellent ion conductivity. Fast and selective ions transport in the designed membrane was verified by using concentration-driven dialysis diffusion tests. The results show that the order of the ion permeation rate through LDH-M is $K^+ > Na^+ > Ca^{2+} > Mg^{2+} > Fe(CN)_6^{3-}$. As expected, the LDH-M shows a lower ion permeation rates (higher ions selectivity) than that of substrate, especially toward $Fe(CN)_6^{3-}$ (Fig. 3b and Supplementary Fig. 6). The prepared LDH-M also demonstrates the selective transport for zincate (Supplementary Fig. 7). This means the LDHs selective layer does play a decisive role in membrane's ion selectivity because of its well-defined interlayer spacing. To confirm the hydroxide ions transport property of the LDH-M,

the hydroxide ion conductivity was detected. As shown in Fig. 3c, the $OH^-$ permeation rate calculated from the slope of the line is similar to that of substrate, indicating the fast $OH^-$ transport capability of LDHs. The hydroxide ion conductivity of LDH-M was investigated and compared with other widely used ion-conducting membranes (Fig. 3d and Supplementary Fig. 8). The conductivity of LDH-M was close to that of substrate, which was on the order of $10^{-2}$ S cm$^{-1}$ and approaching $10^{-1}$ S cm$^{-1}$ in 3 mol L$^{-1}$ NaOH solution. This value is much higher than that of commercialized Nafion membranes (Nafion 115 and Nafion 212) and traditional polybenzimidazoles (PBI) membrane that are commonly used in alkaline systems. To obtain a powerful insight into ions transport behavior in LDH-M, the ion transference number through the membrane was analyzed and calculated by measuring the current–voltage (I–V) curve in a NaOH concentration gradient of 1–3 mol L$^{-1}$ (Fig. 3e). The open-cell voltage ($V_0$) of the device assembled a LDH-M is −17 mV. The $Na^+$ and $OH^-$ transference numbers calculated from Nernst equation[37] for LDH-M is 0.23 and 0.77, respectively (see details

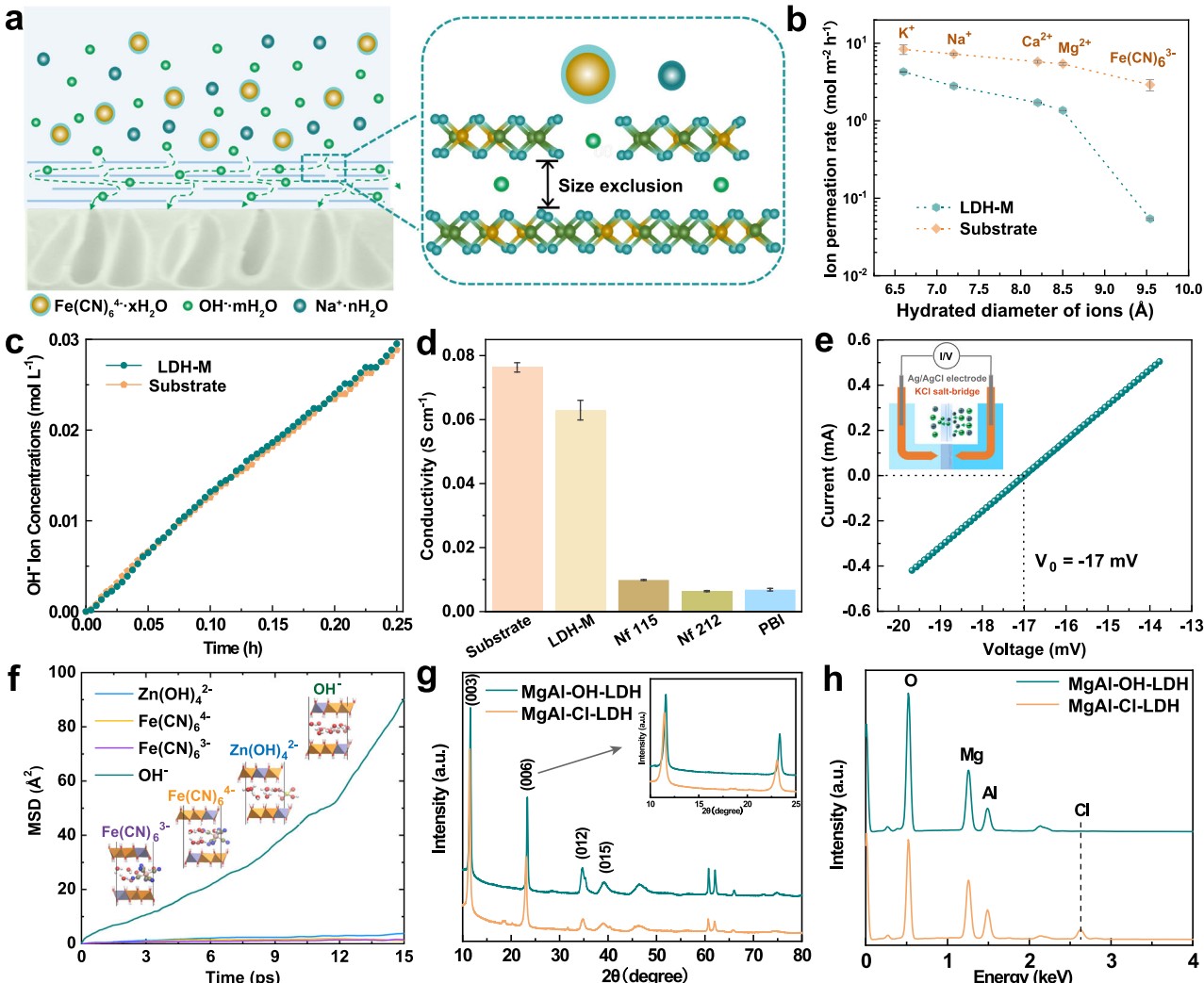

**Fig. 3 Ionic selectivity and conductivity of LDH-M. a** Schematic illustration of the structure of LDH-M and selective ions screening. **b** Selective ions permeation of common salts through LDH-M and substrate. Error bars are standard deviations using at least three measurements from different samples. **c** The permeability of hydroxide ions through LDH-M and substrate. **d** Ionic conductivity of different membranes at 298 K. Error bars are standard deviations using at least three measurements from different samples. **e** The current–voltage (I–V) profile of LDH-M measured in a NaOH concentration gradient of 1–3 mol L$^{-1}$ (The solutions on the left and the right sides of the cells are 1 mol L$^{-1}$ and 3 mol L$^{-1}$ NaOH solutions, respectively). Inset: schematic illustration of I–V testing device. **f** MSD of hydroxide, zincate, ferrocyanide, ferricyanide ions in LDHs at 298 K. **g** XRD patterns of the prepared LDHs (MgAl-Cl-LDH) and NaOH-treated LDHs (MgAl-OH-LDH). Notably, the right shift of the characteristic peak represents the intercalation of hydroxide ions between layers for LDHs treated with NaOH. **h** EDS spectra of MgAl-Cl-LDH and MgAl-OH-LDH, the disappearance of Cl peak demonstrates that Cl$^-$ has been successfully exchanged by OH$^-$.

in methods), indicating that the OH$^-$ acts as the main charging-balancing ions for the designed LDH-M. Compared with the ion transference numbers of commercialized Nafion membranes (Nafion 115 and Nafion 212) and PBI membrane, LDH-M demonstrated high hydroxide ion conductivity (Supplementary Fig. 9). This can be served as the scientific reason behind that the LDH-M is prone to conduct hydroxide ions and LDH-M demonstrates superior ionic conductivity ($7.5 \times 10^{-2}$ S cm$^{-1}$). To identify the effect of LDHs on ions transport behavior of the membrane, we gradually increased the proportion of Nafion binder and regulated the ratio of LDH: Nafion to 4:4, 2:8, 0:1 for comparison. As the proportion of Nafion binder increases (the content of inorganic LDHs nanoparticles decreases), the cross-sectional morphology of the coating layer gradually becomes thinner (Supplementary Fig. 10), the OH$^-$ transference numbers gradually decreased from 0.72 to 0.37 for LDH-M (4:4), LDH-M (2:8), and LDH-Nafion (0:1), respectively (Supplementary

Fig. 11). These proved that the prepared LDH-M (8:2) demonstrate the best performance for OH$^-$ transport, and the LDHs played a main role in ion transport. The fast and selective ions transport in LDHs was further proved by using advanced ab initio molecular dynamics (AIMD) simulation. It is well-known that the greater the slope of the mean square displacement (MSD) illustrated the faster the diffusion. On the basis of MSD of each ions in Fig. 3f, it was demonstrated that the zincate ions (Zn(OH)$_4^{2-}$), ferrocyanide ions (Fe(CN)$_6^{4-}$), and ferricyanide (Fe(CN)$_6^{3-}$) can hardly diffuse into interlayer gallery of LDHs due to their relatively bulky sizes. By contrast, the hydroxide ions (OH$^-$) can easily diffuse into LDHs, and thus a larger MSD value was presented. It indicated that the designed LDHs could effectively screen hydrated Zn(OH)$_4^{2-}$ or Fe(CN)$_6^{4-}$ and simultaneously enable a fast OH$^-$ transport property, and further make the composite membrane very suitable for alkaline-based flow battery application, especially for AZIFB.

Based on the excellent anion exchanging capacity of LDHs, the $Cl^-$ ions in interlayer gallery of MgAl-Cl-LDH can be easily exchanged by hydroxide ions in alkaline environment on account of their weak anchoring strength to LDHs[38]. To confirm that hydroxide ions could pass through the interlayer channel of LDHs, X-ray diffraction (XRD) pattern and energy-dispersive X-ray spectroscopy (EDS) technologies were performed on as-synthesized LDH (MgAl-Cl-LDH) and alkali-treated LDH (MgAl-OH-LDH, MgAl-Cl-LDH nanoparticles treated with $3 \, mol \, L^{-1}$ NaOH at 40 °C for 72 h). As shown in Fig. 3g, the as-synthesized MgAl-Cl-LDH showed strong and symmetric reflections, which are the corresponding (003) and (006) crystal faces, indicating the formation of layered compounds. Compared with MgAl-Cl-LDH, the interlayer spacing of (003) decreases with the interposition of $OH^-$ and the disappearance of Cl element could be observed for MgAl-OH-LDH (Fig. 3h), demonstrating that the $Cl^-$ ions in interlayer gallery have been successfully exchanged by $OH^-$. Furthermore, the exchanged $Cl^-$ can be detected in the supernatant of NaOH solution by using $AgNO_3$ solution titration (Supplementary Fig. 12). Selected area electron diffraction (SAED) data with ETEM were carried out to identify the change in layer spacing of LDHs resulted from the entry of hydroxide ions into the interlayer. SAED results in Fig. 4a, b and corresponding area in Supplementary Fig. 13 show the same tendency as-observed XRD pattern (Fig. 3g) with a decreased interplanar distance for (003) plane on account of the replacement of interlayer anions occurring in alkaline environment. The substituting of $OH^-$ for $Cl^-$ in interlayer gallery could act as a driving force to promote the movement and transport of $OH^-$ and water molecules across the layers, which favors the formation of strong hydrogen bond network. To confirm the aforementioned hydrogen-bond interactions, X-ray photoelectron spectroscopy (XPS) technology was employed (Fig. 4c, d and Supplementary Fig. 14). The O1s spectra can be divided into four characteristic peaks, namely, oxygen atoms bonded to metal (529.9 eV for O1), low coordination oxygen (531.6 eV for O2), hydroxyl groups or their surface-adsorbed oxygens (532.5 eV for O3), and water molecular (533.4 eV for O4)[39]. Compared with MgAl-Cl-LDH, MgAl-OH-LDH demonstrated a higher O3 ratio, indicating that the MgAl-OH-LDH possesses more hydroxyl groups or surface-adsorbed oxygen. These hydroxyl groups or surface-adsorbed oxygens, which stem from the hydrogen bond network among hydroxyl groups in brucite-like host layer, interlayer $OH^-$, and water molecules, can be further confirmed by Fourier transform infrared spectroscopy (FTIR, Supplementary Fig. 15) results. This hydrogen bond network is expected to endow the designed membrane with fast $OH^-$ transport behavior and further high ion conductivity.

**The ions transport mechanism in LDHs**. It was suggested by Sun et al. that the Grotthuss mechanism was responsible for the fast hydroxyl ion conduction in LDHs by performing density functional theory (DFT) simulations[34,35]. However, it remains a challenge for the direct observation of the transport behavior of hydroxide in restricted interlayer gallery of LDHs under operating conditions, and thus confirming the conduction mechanism convincingly. In order to investigate the transport mechanism of $OH^-$ in LDHs layer, ab initio molecular dynamics (AIMD) simulation was explored, which is a powerful tool for providing the dynamic behavior of various species under operating conditions[40–42].

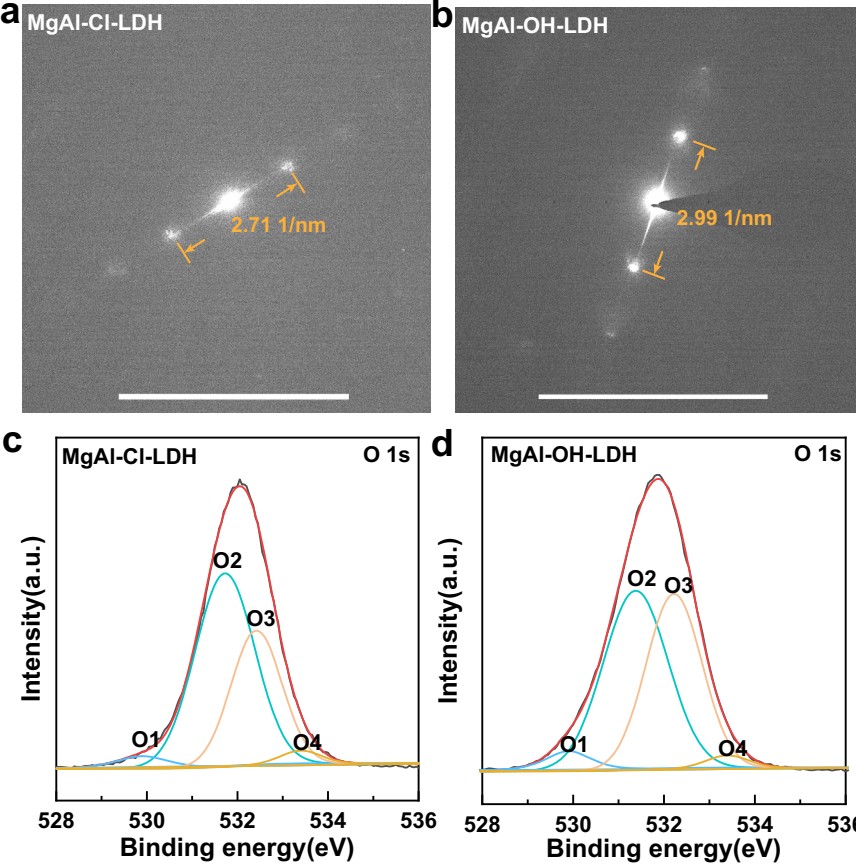

**Fig. 4 Characterizations for LDHs. a, b** The selected area electron diffraction (SAED) of (003) plane for **a** MgAl-Cl-LDH and **b** MgAl-OH-LDH. **c, d** The high resolution O1s core level XPS spectra of **c** MgAl-Cl-LDH and **d** MgAl-OH-LDH. Scale bar: 5 1/nm for **a, b**.

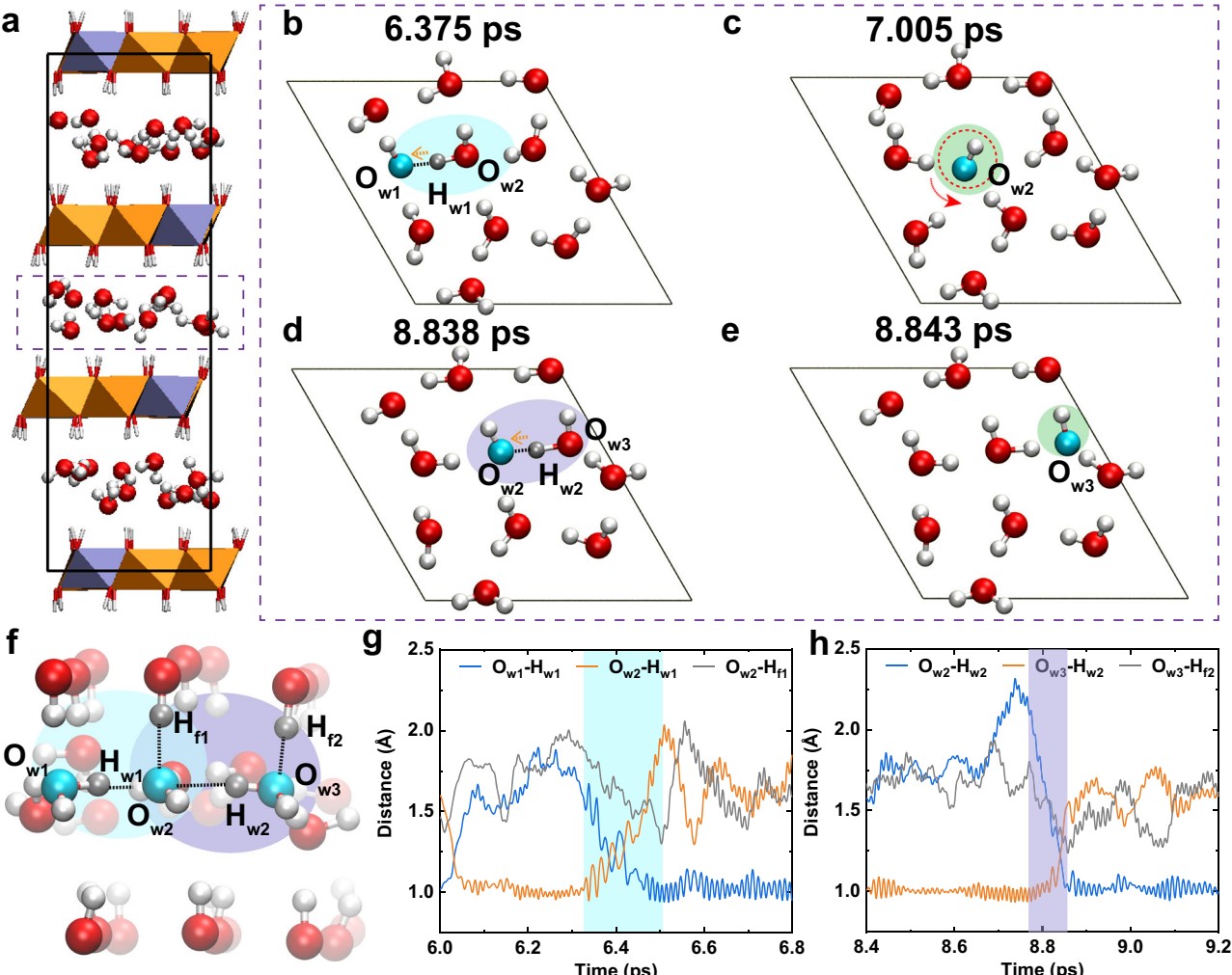

**Fig. 5 Structure, channel, and hydroxide ion conduction of LDHs-based membrane. a** The optimized LDHs structure for simulating the hydroxide ions transport in host layer during the AIMD simulation. **b–e** The snapshots responsible for the conduction of OH⁻ in interlayer gallery of LDHs, which were extracted from the dashed rectangle of **a**. **f** Structure for analyzing the assistance of surface -OH of host layer to the OH⁻ transport with **g–h** the evolution of O–H distances. Mg and Al atoms of brucite-like layer were shown in orange octahedra and ice blue octahedra, respectively. O atoms were shown in red and cyan, and H atoms were shown in white and silver, respectively.

The optimized LDHs model was shown in Fig. 5a and Supplementary Fig. 16, which illustrated that both water molecules and negative OH⁻ ions were located in interlayer gallery. In order to shed light upon the transport behavior of OH⁻, the selected chronological snapshots during the AIMD simulation (Supplementary Movie 1) were depicted in Fig. 5b–e, which indicated that the transport of OH⁻ could be considered as a reverse transport of proton in nano-confined LDHs layers. Interestingly, similar to the case of proton, the OH⁻ group can transmit through both Grotthuss mechanism (proton hopping)[43,44] and vehicular mechanism (standard diffusion)[45–47]. In detail, the hydroxide ion was initially located at the $O_{w1}H^-$ position (Fig. 5b); while as time processes, the proton $H_{w1}$ (colored in silver) hopped from $O_{w2}$ atom to $O_{w1}$ atom via the Grotthuss mechanism, leading to the transport of hydroxide ion from $O_{w1}H^-$ to $O_{w2}H^-$ (Fig. 5b, c, colored in cyan shadow). Subsequently, the $O_{w2}H^-$ groups translate and rotate inside the LDHs layer and transport through the vehicular mechanism (Fig. 5c, d). Similarly, the transport of OH⁻ sequentially conducted from $O_{w2}H^-$ to $O_{w3}H^-$ (Fig. 5d, e, colored in purple shadow) via the backward transport of proton ($H_{w2}$). Apparently, the synergies of Grotthuss mechanism and vehicular mechanism will contribute to the fast transport of OH⁻ in the

LDHs layers. Furthermore, we also calculated the mean square displacement (MSD) of all O atoms in both water and OH⁻ for comparison, which can qualitatively reflect the contribution of vehicular mechanism to some extent (Supplementary Fig. 17). As a result, the fast transport of OH⁻ in the LDHs layers was mainly contributed by the Grotthuss mechanism.

On the other hand, the surface -OH groups (e.g., $-OH_{f1}$ and $-OH_{f2}$ in Fig. 5f) of LDHs layers played a crucial role in assisting the transport of hydroxide ions as well. As depicted in Fig. 5b, the $H_{w1}$ proton was initially located at the $O_{w2}$ atom and the hydroxide ion existed as $O_{w1}H^-$. During the transport process of $O_{w1}H^-$ to $O_{w2}H^-$, the distance of $O_{w2}-H_{f1}$ decreased as well as forming hydrogen bond, demonstrating that surface -OH groups ($-OH_{f1}$) effectively enhanced the transport ability of $H_{w1}$ (Fig. 5g). The similar effect of another surface -OH ($-OH_{f2}$) was observed at ca. 8.84 ps which assisted the transport of $H_{w2}$ from $O_{w3}$ to $O_{w2}$ (Fig. 5h), and hence facilitating the transport of $O_{w2}H^-$ to $O_{w3}H^-$ (Fig. 5d, e). Overall, the surface -OH groups of LDHs layer could strongly promote the proton dissociation and transfer away from one water molecule to the original OH⁻, yielding an extremely high OH⁻ conductivity.

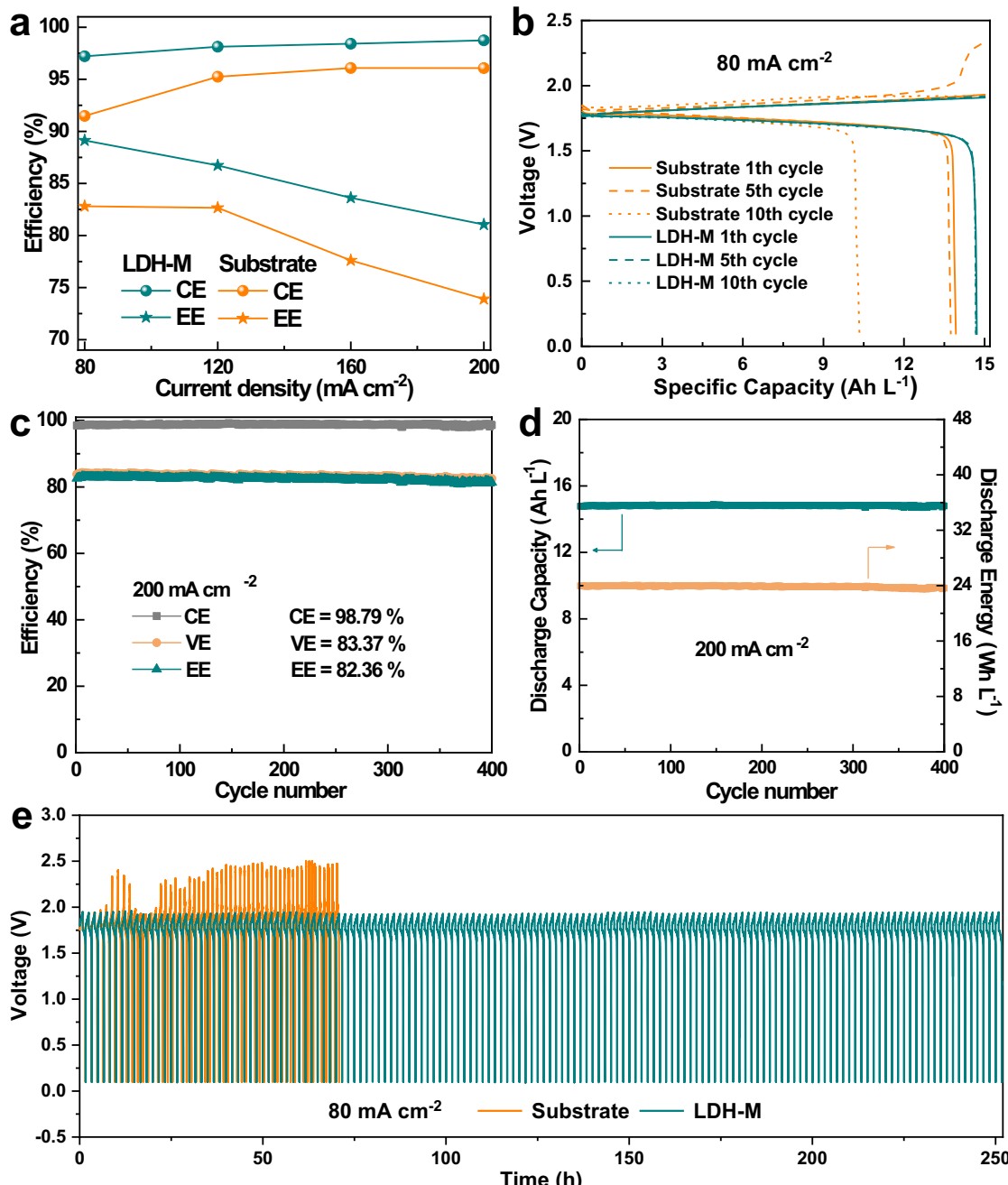

**Fig. 6 Electrochemical performance of the AZIFB employing LDH-M and substrate. a** Rate performance as the current density ranging from 80 to 200 mA cm$^{-2}$. **b** The voltage profiles in the 1th, 5th, 10th cycles for AZIFB. **c** The cycling performance of AZIFB assembled with a LDH-M at 200 mA cm$^{-2}$. **d** Corresponding discharge capacity and discharge energy for each cycle at 200 mA cm$^{-2}$. **e** The voltage–time profiles at a current density of 80 mA cm$^{-2}$.

**LDH-M for efficient and stable flow battery application.** As mentioned above, LDH nanoparticles are selected due to their high ion selectivity, superb hydroxide ion conductivity, and robust stability in alkaline environment. As a proof of concept, the LDH-M membrane was investigated in AZIFB. The rate performances of the AZIFB assembled with LDH-M and substrate were shown in Fig. 6a. As the current density increased from 80 to 200 mA cm$^{-2}$, an increased coulombic efficiency (CE) was observed for AZIFB with LDH-M and substrate, while the voltage efficiency (VE) and EE slightly decreased due to the increased Ohmic polarization and electrochemical polarization for the battery. An AZIFB with LDH-M delivered a significant improvement of CE and similar VE compared to the battery with

substrate. Compared with the membranes prepared from different ratio of LDHs/Nafion, LDH-M still demonstrate the highest EE (Supplementary Fig. 18). This result is attributed to the coating layer of LDHs that can enhance the ion selectivity of the membrane and keep high hydroxide ion conductivity. Lower contact angle indicates better electrolyte wettability for LDH-M thus bringing high ion conductivity and high VE (Supplementary Fig. 19). Figure 6b shows the charge–discharge curves of AZIFB with LDH-M and substrate at a current density of 80 mA cm$^{-2}$. An AZIFB assembled with a LDH-M delivered an initial CE of 98.0%, a VE of 92.4%, and a cycling stability with a stable discharge capacity of about 15 Ah L$^{-1}$ for nearly 150 cycles (Supplementary Fig. 20). By contrast, the discharge capacity of

the battery assembled with substrate decreased gradually during cycling due to the crossover of active materials ($Zn(OH)_4^{2-}$ and $Fe(CN)_6^{4-}/Fe(CN)_6^{3-}$) (Supplementary Fig. 21), leading to a self-discharge and further decreasing the discharge capacity of the battery. Additionally, the crossover of the electrolyte would decrease the concentration of the active materials in both positive and negative half-cells. Thus, a serious concentration polarization of the battery can be observed as evidenced by the sharply increased voltage at the end of charging of the battery (Fig. 6e). In contrast, LDH-M demonstrated a much lower permeation rate for active materials, which endows the AZIFB with a stable voltage–time profile. Even at a high current density of 200 mA cm$^{-2}$, the AZIFB assembled with a LDH-M still afforded a stable performance for nearly 400 cycles, maintaining an average EE of 82.36% (Fig. 6c), which is much higher than that of the battery with Nafion 212, Nafion 115, PBI and porous substrate (Supplementary Fig. 22). The AZIFB demonstrated a very competitive performance among recently reported zinc-based flow batteries (Supplementary Table 1)[48]. Furthermore, the battery assembled with LDH-M showed a stable discharge capacity of 14.81 Ah L$^{-1}$ and a discharge energy of 23.87 Wh L$^{-1}$ for 400 cycles (Fig. 6d). To identify the mechanical stability of LDH-M, the mechanical stability LDH-M after more than 400 cycles battery test at the current density of 200 mA cm$^{-2}$ (LDH-M-Cycle) was tested, the almost similar values of mechanical performance for LDH-M-Cycle indicated the good mechanical stability of LDH-M (Supplementary Table 2). Furthermore, the structures and cross-section and surface morphologies of the LDH-M-Cycle were characterized by XRD and FE-SEM, respectively. The remained LDHs flake layer on the porous substrate confirming the stability of the LDHs (Supplementary Fig. 23). The high working current density together with the high open-circuit voltage (OCV) (Supplementary Fig. 24a) can thus endow the battery with a high-power density (Supplementary Fig. 24b). Taken together, the as-designed composite membrane with LDHs layer, ensuring a perfect combination of high ion selectivity and conductivity, offers a promising candidate for alkaline-based flow batteries.

## Discussion

In summary, a LDHs-based membrane was designed and introduced into flow battery application. The high ionic selectivity and hydroxide ion conductivity of the membrane were identified in detail. The experimental results show that LDHs-based membrane demonstrate a high selectivity for different ions, especially for $Fe(CN)_6^{3-}$. Hydroxide ionic conductivity was confirmed for LDH-M, which demonstrated a superior ionic conductivity of $7.5 \times 10^{-2}$ S cm$^{-1}$. It is noteworthy that the hydroxide ions transporting through LDH via vehicular & Grotthuss mechanisms can be identified according to the AIMD simulations. Furthermore, the results identified that surface -OH groups of LDHs layer could assist the conduction of OH$^-$ by promoting proton transfer away from one water molecule to the original OH$^-$, yielding extremely high ionic conductivity for LDH-M. As a proof of concept, an AZIFB assembled with a LDH-M exhibits a high CE at different current densities, since the ordered gallery height for LDHs can efficiently impede active species while transferring the charge carrier OH$^-$. More importantly, an AZIFB with a LDH-M membrane can maintain an EE of above 82% and a stable cycling performance for more than 400 cycles at a current density of 200 mA cm$^{-2}$, demonstrating a competitive performance among the zinc-based flow batteries. These findings open a new pathway for the development of diverse membranes with multifunctional ion channel for efficient ion separation, energy reservation, and power generation.

## Methods

**Materials.** Poly(ether ether ketone) (PEEK) and poly(ether sulfone) (PES) were provided by Changchun Jilin University Special Plastic Engineering Research. Sulfonated poly(ether ether ketone) (SPEEK) was prepared by direct sulfonation of PEEK with sulfuric acid (98%) at 40 ºC for 3 h. Potassium hydroxide, N, N-dimethylacetamide (DMAc) were purchased from Tianjin Damao Chemical Reagent Factory. Magnesium chloride, aluminum chloride, sodium hydroxide, sodium ferrocyanide, potassium ferricyanide, and zinc oxide were bought from Kermel Chemical Reagent Factory. Potassium chloride (99.5%), sodium chloride (99.5%), and calcium chloride dihydrate were purchased from Aladdin, Shanghai. These reagents were supplied with analytical grade (AR).

**Synthesis of LDHs.** MgAl-Cl-LDH nanoparticles were prepared using the co-precipitation and hydrothermal treatment method[36]. To prepare MgAl-Cl-LDH nanoparticles, a mixture of 0.3 mol L$^{-1}$ MgCl$_2$ and 0.1 mol L$^{-1}$ AlCl$_3$ solution (10 mL) was quickly added to 40 mL of 0.15 mol L$^{-1}$ NaOH solution with 10 min stirring, then the LDH slurry was separated by centrifugation and washed with deionized water. The dispersed solution was transferred into a stainless Teflon-lined stainless steel autoclave for hydrothermal reaction at 100 ºC for 16 h. Then a transparent, homogenous suspension containing MgAl-Cl-LDH nanoparticles was obtained, followed by filtration separation, deionized water washing, and freeze drying for use. MgAl-OH-LDH was prepared by soaking MgAl-Cl-LDH in 3 mol L$^{-1}$ NaOH solution at 40 ºC for 72 h, then freeze drying for use.

**Preparation of porous support (substrate).** The poly(ether sulfone) (PES) porous membrane was prepared by the phase inversion method. The polymers (10 wt.% SPEEK in the polymer and 35 wt.% for polymer concentration) were first dissolved in DMAc, then casted the above solution onto a clean glass plate at room temperature with humidity less than 30%. Afterwards, the plate was immersed into water to form the PES membrane.

**Preparation of LDHs-coated composite membrane.** The as-prepared LDHs nanoparticles were first dispersed in ethanol and sonicated for 4 h using a sonic bath (KQ5200) to form LDH nanosheets. Then a certain amount of 1.0 wt.% Nafion solution served as binder was added into the above suspension (the mass ratio of LDHs/Nafion is 8/2). The 1.0 wt.% Nafion solution was prepared by diluting 5.0 wt.% Nafion dispersion (Dupont, D-520) with isopropanol (IPA). The resulted suspension was sonicated for another 4 h and formed the LDHs dispersion (the dispersion concentration is 40 mg mL$^{-1}$). Then 1 mL of above LDHs dispersion was slowly and evenly sprayed onto the prepared porous support, forming the LDH-M.

**LDHs characterization.** The high resolution diffraction of LDHs were recorded on the Titan Themis G3 ETEM (Thermo Scientific Company) at 80 kV with a Cs corrector for parallel imaging (CEOS GmbH). High-resolution transmission electron microscopy (HRTEM, JEM-2100) was conducted to characterize the morphology of the LDHs. X-ray diffractometer (D8 ADVANCE ECO; RIGAKU, Japan) was used to detect the powder XRD patterns of MgAl-Cl-LDH and MgAl-OH-LDH nanoparticles, which has a monochromatic Cu-Kα radiation source at 40 kV and 40 mA and scan rate of 10º min$^{-1}$. The disappearance of chlorine element was detected using energy-dispersive X-ray spectroscopy (EDS). X-ray photoelectron spectroscopy (XPS) was conducted on a Thermo ESCALAB 250XI at 150 W (Al Kαradiation, 1486.6 eV).

**Membrane characterization.** Field-emission scanning electron microscopy (FE-SEM, JEOL 6360LV, Japan) and energy-dispersive X-ray spectroscopy (EDS) were used to detect the morphologies of prepared membranes. The membranes were treated by breaking them in liquid nitrogen and sprayed with gold to obtain the cross-sections before imaging. The thickness of the LDHs flaker layer on the substrate after cycling (400 cycles, 200 mA cm$^{-2}$) was measured by FE-SEM, the membrane was collected by disassembling the battery and washed with ultra-pure water. The contact angle meter (POWEREACH, China) was used to clarify the wettability between membranes and the electrolyte (3 mol L$^{-1}$ NaOH). X-ray diffractometer (D8 ADVANCE ECO; RIGAKU, Japan) with a monochromatic Cu-Kα radiation source at 40 kV and 40 mA and scan rate of 10º min$^{-1}$ was used to identify the stability of LDHs on the LDH-M.

**Membrane conductivity.** Electrochemical impedance spectroscopy (EIS) testing station (Solartron SI 1260 and SI 1287) was used to test the membrane conductivity as reported[42]. The range of frequency was set to be 1–100 KHz. Pieces of membrane were sandwiched between two round titanium plates with 1.5 cm in diameter. The membrane conductivity was calculated as the following equation.

$$\sigma = \frac{L}{R \times A} \qquad (1)$$

$\sigma$ (S cm$^{-1}$), $L$ (cm), $R$ ($\Omega$), and $A$ (cm$^2$) are the conductivity, the thickness, the resistance, and the effective area of the membrane, respectively.

**The ionic transport properties**. The ionic transport properties of different membranes were investigated using Gamry Interface 3000. The current–voltage (I–V) profile was recorded when the membrane was sandwiched between two cells soaking with a gradient of 1–3 mol L$^{-1}$ NaOH solution. Two Ag/AgCl reference electrodes filled with saturated KCl solution and two salt bridges filled with saturated KCl solution were employed to eliminate the potential drop. Thus, the open-cell voltage of the device ($V_o$) is equal to the value of diffusion potential ($V_d$) resulted from the NaOH concentration gradient, which can be calculated as the following equation.

$$V_o = V_d = \frac{RT}{F}(t_{Na^+} - t_{OH^-})\ln(\Delta) \qquad (2)$$

R, T, F, $t_{Na^+}$, $t_{OH^-}$, and $\Delta$ are the gas constant, temperature, faraday constant, Na$^+$ transference number, OH$^-$ transference number, and activity gradient (the mean ion activity coefficient was considered since the concentration of NaOH solution is high), respectively.

**The permeability of different ions and hydroxide ions**. The permeability of different ions was tested by a diffusion cell. The feed solution: 1 mol L$^{-1}$ salt solution in ultra-pure water, including KCl, NaCl, CaCl$_2$, MgCl$_2$, and K$_3$Fe(CN)$_6$ with a varied hydrated diameter of cations or anions, respectively. The permeate side: ultra-pure water. The ion concentration of the diffusion side was then calculated as following equation[49].

$$\Lambda_m = \frac{\kappa}{c} \qquad (3)$$

$\kappa$ is the conductivity of solution in the diffusion side, and $c$ is the ion concentration. The conductivity of the diffusion side with a certain concentration was first measured. Then the slope of the plot of conductivity and concentration was the molar conductivity $\Lambda_m$ of metal chloride in the diffusion side.

The permeation of zincate ion (Zn(OH)$_4^{2-}$) through the prepared membranes were determined by a diffusion cell. The feed solution: 0.4 mol L$^{-1}$ Na$_2$Zn(OH)$_4$ in 3 mol L$^{-1}$ NaOH. The permeate side: 0.4 mol L$^{-1}$ Na$_2$SO$_4$ in 3 mol L$^{-1}$ NaOH to equalize the ionic strengths and minimize the osmotic pressure effects. The effective area of the membrane was 9 cm$^2$. 3 mL samples of the solution were collected from the permeate side at a regular time interval. The Zn(OH)$_4^{2-}$ concentration of the samples was detected using an inductively coupled plasma mass spectrometry (ICP-MS).

The permeability of hydroxyl ions across the membranes was also tested using an osmosis cell, and the hydroxyl ions concentration was measured by Mettler Toledo pH meter. The feeding solution and the diffusion side were using 3 mol L$^{-1}$ NaOH and ulta-pure water, respectively.

**The porosity of the membrane**. The membrane was immersed in water for 24 h and then weighed after wiping off the water on the surface with filter paper. Subsequently, weighed the membrane after fully dried it in a vacuum oven. The porosity of the membrane was calculated as follows:

$$\varepsilon = \frac{M_w - M_d}{\rho \times S \times l} \qquad (4)$$

where $M_w$ and $M_d$ are the mass of the wet and dried membrane, respectively; $\rho$ is the density of water at room temperature; $S$ is the surface area of the dried membrane; $l$ is the thickness of the dried membrane.

**Mechanical property**. The mechanical stability of the membranes was measured by tensile test, including tensile strength, breaking stress, elongation at break. The membrane is cut into a size of 1 × 7 cm, and 5 replicates are prepared for each sample. Then measure the thickness of the membrane with a spiral micrometer. Fix the sample on the universal testing machine, the left and right clamping width is 1 cm each, the stretch gauge length is 5 cm, and the stretch rate is 5 mm/min.

**Electrochemical performance of the alkaline zinc–iron flow battery**. The AZIFB was assembled by sandwiching the prepared membrane between two carbon felt electrodes (3 × 3 cm$^2$), clamped by two graphite plates. The thickness of the carbon felt electrode is 5 mm and the compression ratio of the carbon felt electrode is 1.43. The composite membrane with LDHs was facing the positive side of the battery. The negative and positive electrolytes were 40 mL 0.4 mol L$^{-1}$ Zn(OH)$_4^{2-}$ + 3 mol L$^{-1}$ OH$^-$ and 40 mL 0.8 mol L$^{-1}$ Fe(CN)$_6^{4-}$ + 3 mol L$^{-1}$ OH$^-$, respectively. The electrolytes were cyclically pumped through the corresponding electrodes in airtight pipelines. Charge–discharge tests were carried out on ArbinBT 2000 at different current densities (80–200 mA cm$^{-2}$). Constant charge capacity was controlled through time cutoff (50 min at 80 mA cm$^{-2}$ and 20 min at 200 mA cm$^{-2}$) during charge process, while the discharge process was ended with a cut-off voltage of 0.1 V. The polarization curves were conducted by charging to 90%, 60%, 30% SOC at 40 mA cm$^{-2}$ and discharging at different current densities, respectively.

**Models for LDHs**. Mg$_2$Al(OH)$_6$Cl·2H$_2$O (model-0) is a commonly layered double hydroxide (MgAl-Cl-LDH). The MgAl-Cl-LDH was constructed from the 3 × 3 × 1 supercells of brucite (Mg(OH)$_2$) but substituting 1/3 Mg$^{2+}$ with Al$^{3+}$, the

induced positive charge after substitution was balanced by interlayer Cl$^-$ with the accompany of water molecules, and fully relaxed subsequently. The unit cell parameters for MgAl-Cl-LDH model were referred to Wang's work[50]. After the optimization, the Cl$^-$ was all replaced by the OH$^-$ (MgAl-OH-LDH, model-1, Fig. 5a) to investigate the conductivity of OH$^-$ in LDHs layer. It should be noted that the theoretical simulations were explored under hydrated environment, and hence each layer of MgAl-OH-LDH structure (model-1) was composed of 3 hydroxide anions and 8 water molecules. The number of water molecules was calculated according to water density at atmospheric pressure (1 g cm$^{-3}$) by multiplying the free volume (enclosed by the Connolly surface with the Connolly radius set as 1.0 Å). In order to keep the charge neutrality of the simulated LDH system, the number of OH$^-$ was determined based on the number of Mg$^{2+}$ replaced by Al$^{3+}$. Additionally, to further study the selectivity of LDHs layer, the Zn(OH)$_4^{2-}$, Fe(CN)$_6^{3-}$, and Fe(CN)$_6^{4-}$ replaced respective 2, 3, and 4 Cl$^-$ on the basis of model-0 to obtain model-2, model-3, and model-4, respectively. The four models were accurately optimized using advanced periodic density functional theory (DFT). The optimized models were all shown in Supplementary Fig. 16.

**Ab initio molecular dynamics and DFT optimization**. The optimization as well as ab initio molecular dynamics (AIMD) simulation were carried out with the mixed Gaussian plane wave scheme using the CP2K package (version 4.1)[51–53]. Then, the Perdew, Burke, Ernzrhof (PBE) exchange-correlation functional[54], and the DZVP-MOLOPT-SR basis set with Goedecker–Teter–Hutter (GTH) pseudo potentials[55] were used. During the calculation, the Grimme D3 correction[56] with zero damping was applied to account for the dispersion interactions, as well as the plane wave cutoff energy and relative cutoff were 650 Ry and 60 Ry, respectively. The four structures (model-1, model-2, model-3, and model-4) were relaxed before performing AIMD simulation. During the AIMD process, a 20 ps simulation with a time step of 0.5 fs and a coupling time constant of 100 fs was performed in the NVT ensemble at 298 K, and controlled by the Nosé–Hoover thermostat[57]. The trajectories were recorded every step to analyze the mean square displacement.

**Reporting summary**. Further information on research design is available in the Nature Research Reporting Summary linked to this article.

## Data availability

The data that support the findings of this study are available from the corresponding author upon request.

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

## Acknowledgements

The authors greatly acknowledge the financial support from NSFC (22078313, 21908214, 21925804), CAS Engineering Laboratory for Electrochemical Energy Storage, CAS interdisciplinary innovation Team (Grant No. JCTD-2018-10) and Liaoning Revitalization Talents Program (XLYC1802050), DICP funding (DICP ZZBS201814), Youth Innovation Promotion Association CAS (2019182), and DNL Cooperation Found, CAS (DNL201910).

## Author contributions

J.H. performed the experiment and analyzed the data. J.H., X.M.T., Z.Q.L., Z.Z.Y., and A. M.Z. discussed and designed the calculation. X.M. Tang and Z.Q.L. performed the calculation. J.H., Q.D., H.M.Z., Z.Z.Y., and X.F.L. participated in project planning and discussions of the results. All authors reviewed the manuscript.

## Competing interests

The authors declare no competing interests.
