## [Peer Review File · Nature Communications]

REVIEWER COMMENTS

Reviewer #1 (Remarks to the Author):

The manuscript offered by Li and co-workers describes the use of layered double hydroxide (LDH) materials as fast hydroxide conductors that, when incorporated into a membrane, also exhibit selective ion transporting properties. The foundations for this work have already been described by Sun, et al. in two preceding publications: *Sci. Adv.* 3, e1602629 (2017) and *Mater. Horiz.* 6, 2087-2093 (2019). Specifically, LDHs had already shown superionic conductivity for hydroxide, setting an inversion of precedent for more typical fast transport of H⁺. Mechanistic insights into the transport mechanisms, local to the gallery between the sheets and longer range via tortuosity between particulates had also been described deeply therein. Where this manuscript endeavors to go further is to show that ion-conducting LDHs can be composited with Nafion polymers and used as a flow battery membrane for select aqueous cell chemistries, which depend on the membrane's blocking character for specific ions in specific electrolytes. In this way, the results presented herein may not significantly advance the field beyond the state-of-the-art. More specifically, demonstration of a Zn/Fe flow cell operating at high current density is not in itself substantive enough to motivate publication, particularly as the principal technical and scientific conclusions related to the LDH-based membrane and its properties are not yet fully developed, consistent, or supported by the theory and the experiments, which is described in more detail below. Consequently, I cannot recommend this work in its current state for publication in *Nature Communications*.

In their introduction, the authors appear to leave out the most well-studied size-sieving membranes for energy storage: microporous polymers. If we confine the discussion to flow batteries, then it would be appropriate to include citations to: *Nature Materials* 19, 195–202 (2020); *Joule* 3, 2968–2985 (2019); *Adv. Energy Mater.* 6, 1600517 (2016); *Nano Lett.* 15, 5724–5729 (2015). Some of these are also hydroxide ion conductors and have been used as selective/blocking membranes in aqueous Zn/Fe batteries, which is particularly relevant to the discussion of the state-of-the-art in this manuscript.

In highlighting the performance of devices incorporating their membranes, they make bold claims as to the current density supported (200 mA/cm²) as world-leading and a breakthrough enabled by the membrane. However, they do not provide references for comparison to what the leading edge might have been prior to this work. The authors will need to provide well researched documentation for this claim by citing the previous state of the art.

The authors are asked to differentiate better for the reader the theoretical results presented from those already described for superionic transport of hydroxide in LDHs.

The authors seem overly confident that interlayer transport of ions through the gallery is the main mechanism governing the transport behaviors of different ions in the membrane, e.g., in Figure 3e and elsewhere. They neglect contributions arising from the LDH-Nafion composite architecture, which results in a multiplicity of interfaces, arrangements of LDH particles in the composite, etc. In this way, their interpretation of the transport behaviors is too premature. To resolve this, it is necessary to provide evidence that the mechanism proposed and explored is dominant. This could involve, e.g., investigations of the properties of membranes with variable volume fraction of the LDH particles in the composite. Here, the authors would need to link structural outcomes of the compositing process with variable LDH volume fraction and the ensuing transport properties (conductivity, blocking ability, etc.). For example, at low volume fraction of LDH, the membrane should behave like Nafion. At high volume fraction, we recover the behavior described. In the intermediate range, it should be possible to assess the relative contributions of interfacial transport vs. within-LDH transport, perhaps also evidencing the volume fraction of LDH leading to percolation of those domains within the composite membrane.

Linking this to theory is also important. For example, Nafion is a fluoropolymer. It could be that the interface between it and LDH is poorly defined, resulting in gaps between the polymer and the particles; these gaps often contribute to the transport behaviors observed. Alternatively, depending on the surface chemistry of the LDH, the interface can densify, forcing ions to diffuse within the interlayers, but experiencing high resistance as you cross between the interlayer, the densified Nafion, and onto the next LDH. The authors are further advised to consider the microstructural features of hydrated Nafion in the composite on the membrane transport properties. Thus, while the theoretical and characterization methods performed thus far are the logical extension from what was done previously, they focus too narrowly on diffusivity within interlayer, which may only be a minor contributor to the overall transport mechanism in a composite with Nafion.

It would be useful to evaluate the [hydroxide]-dependent conductivity of the membrane. This also serves to distinguish the properties of the membrane for alkaline flow batteries relative to Nafion. Currently, the data presented in Figure 3 does not yet establish these membranes as fast ion conductors for hydroxide relative to the state of the art. Figure 3d, in particular, may just as well arise from a defective membrane (i.e., with an aberration or tear). To convince the reader, it is necessary to evaluate the conductivity of the membrane with variable area, thickness, etc. To convincingly demonstrate there is not a tear or other aberration, it is necessary to measure the Gurley number. To convincingly demonstrate the membrane can be handled without being damaged, the authors would need to evaluate the mechanical properties of the membrane, as is typical for Nafion and similar membranes for flow batteries and fuel cells.

The Zn/Fe redox-flow cells have been demonstrated with low volumetric capacity (~ 15 Ah/L) yet comparable volumetric energy density (~ 25 Wh/L) to the more recognizable vanadium redox-flow batteries. If the membrane is selective as purported, it should be possible to increase those significantly and elevate the potential for impact. Many emerging flow battery chemistries can go even higher on those fronts.

The authors are respectfully asked to avoid using nebulous terms such as "Base" when describing any control membranes in the flow cell work. Just specify what "Base" actually means. In particular, the authors should provide to the reader as assessment of the flow cells where Nafion is used (i.e., as a negative control where the LDH is not featured in the membrane) and where a state-of-the-art membrane (e.g., PBI as a positive control) is used. It appears from their previous work that this has been done before and could provide the reader here with useful information needed to understand the progress made and enabled by the LDH component in the membrane.

The authors are asked to consider and relate to the reader more directly what specific problem or application in grid storage these flow battery cells might be addressing.

Reviewer #2 (Remarks to the Author):

This work reports a LDH membrane having a high OH⁻ conductivity and high ion selectivity and its use for an alkaline flow battery. The new insight that this work provides would be fast OH⁻ transport via Grotthous mechanism based on MD simulation, which is facilitated by the LDH structure. The performances of the membrane are interesting and meaningful in this technology sector, but the physics regarding the ion, and mass transport need to be further clarified.

1) The structure of base membrane should be described in more detail. Why is PES selected as a base? Porosity and pore size distribution would be needed. Are the finger-like macropores the major channel for mass transport in the base membrane? If so, is there any problem of inhomogeneous flux?. The

interface between LDH and base membrane is strong enough? Is the interfacial stability evaluated?

2) Although the authors state that the LDH particles are well aligned in the membrane (30micron), many interstitial pores appear. What is the porosity and poresize distribution of the LDH layer? Can the contribution from the pore be neglected in the conductivity analysis? Or is the membrane dense due to the Nafion binder? Since the nano channels of a single particle are not aligned with each other (It would be not possible), the ions should find their way from the edge of one particle to that of nearby particle. Therefore, the interstitial pore may have its effect on transport. In the case that interstitial pores are connected with a lower resistance, molecules would pass through the pores instead of nanochannel. Compression (densification) of LDH membrane would show the effect of the interstitial pores.

3) Is the selectivity of LDH membrane higher than Nafion 212, PBI, 115 membrane? Compare the transference number for LDH, Nafion and PBI also.

4) For MD simulation, how come the numbers of water and OH⁻ molecule are defined. Is it based on experiment or assumption? Is electric field applied to the set? The existence of Grotthous mechanism is feasible. However, it is not clear how large its contribution in comparison with vehicular mechanism. Like nanocapillary action of water flow in graphene nanosheet (Science, 2014, 343, 752-754), capillary flow can happen.

5) In comparison of base and LDH-base membrane, base cell showed slightly higher voltage efficiency. Why? the ohmic resistance of membrane should be larger for LDH-base membrane because LDH layer is added to the base. In comparison with Nafion or PBI membrane, does LDH membrane have higher coulombic and energy efficiency?

Reviewer #3 (Remarks to the Author):

This work reported a novel layered double hydroxides (LDHs) composite membrane with a high selectivity and superb hydroxide ion conductivity for alkaline-based flow batteries. The transport behavior of OH⁻ in restricted interlayer gallery of LDHs was clearly presented and clarified. The contribution created the application of LDHs nanomaterials in alkaline-based flow batteries and achieved very encouraging performance ever reported. This is an impressive work in the field of size-based separation materials and alkaline based batteries. The work will provide very important information on understanding of transport mechanism of OH⁻ in LDHs nanomaterials, as well as on both 2D materials and energy storage devices. Therefore, I recommend accepting this paper in Nature Communications after considering the following minor comments:

1. The anti-alkaline stability of layered double hydroxide membrane (LDH-M) under the complex environment of electric field and strong alkali after battery cycling should be verified by XRD results though the stability of LDHs in alkaline environment was presented (Figure 2g).
2. The charge-discharge profiles at the end of charging for Base appears a sudden arise in Figure 6b, the authors need to comment on this.
3. From the area resistance results (Figure 3d), LDH-M seem to present a higher area resistance, which should lead to lower voltage efficiencies, but higher VEs and EEs are achieved in the battery tests (Figure 6a). Pls clarify this?
4. For the DFT calculation part, the function and basis set should be provided, and the selectivity of Fe(CN)₆³⁻ for DFT calculation as mentioned should be included in the SI. Also, please explain the reason why including model-2 and model-3 in this simulation.
5. The experimental details of assembling the flow battery should be provided.

6. The format through the manuscript should be carefully checked and improved, e.g., the Ref. No oscillated before and after punctuations.

Response to Reviewer 1:

The manuscript offered by Li and co-workers describes the use of layered double hydroxide (LDH) materials as fast hydroxide conductors that, when incorporated into a membrane, also exhibit selective ion transporting properties. The foundations for this work have already been described by Sun, et al. in two preceding publications: *Sci. Adv.* 3, e1602629 (2017) and *Mater. Horiz.* 6, 2087-2093 (2019). Specifically, LDHs had already shown superionic conductivity for hydroxide, setting an inversion of precedent for more typical fast transport of H^+ . Mechanistic insights into the transport mechanisms, local to the gallery between the sheets and longer range via tortuosity between particulates had also been described deeply therein. Where this manuscript endeavors to go further is to show that ion-conducting LDHs can be composited with Nafion polymers and used as a flow battery membrane for select aqueous cell chemistries, which depend on the membrane's blocking character for specific ions in specific electrolytes. In this way, the results presented herein may not significantly advance the field beyond the state-of-the-art. More specifically, demonstration of a Zn/Fe flow cell operating at high current density is not in itself substantive enough to motivate publication, particularly as the principal technical and scientific conclusions related to the LDH-based membrane and its properties are not yet fully developed, consistent, or supported by the theory and the experiments, which is described in more detail below. Consequently, I cannot recommend this work in its current state for publication in *Nature Communications*.

Response: We really appreciate the reviewer's efforts on our work and thank the reviewer for providing insightful comments to further improve the quality of our manuscript. Regarding these comments, we would like to clarify below the reviewer's concerns, which we hope to make this manuscript powerful to the high standard of *Nature Communications*. Critically different from the previous works (*Sci. Adv.* 3, e1602629 (2017); *Mater. Horiz.* 6, 2087-2093 (2019)), our work had made very important achievements as follows:

(1) Actually, energy storage devices are of very important for wide application of renewable energies. And in this paper for the first time, we have introduced layered double hydroxide (LDH) nanomaterial in alkaline based flow battery systems by using their specific OH^- transport behavior and size exclusion of well-defined layered channels, which is of great significance to broaden the application field of LDHs, which we think is one of the most important missions for materials.

(2) Very importantly, *ab initio* molecular dynamics (AIMD) simulation was performed in this work to investigate the ions selective transport behavior, which provided a direct observation and evidence of the dynamic behaviors for hydroxide as well as other kind of species in restricted interlayer gallery of LDHs, which could represent the realistic operating conditions to some extent.

(3) To note that: the LDHs were introduced into the porous substrate to form the composite membrane, of which a minor amount of Nafion here was only acted as the binder and does not play a decisive role in the overall performance of the membrane. Demonstration of a Zn/Fe flow battery opens a new pathway for the development of diverse membranes with multifunctional ion channel for efficient ion separation and energy storage.

Therefore, we believe that this paper could provide very important information both for membranes separation, energy storage and two-dimensional (2D) materials.

Comments to the Author

(1) In their introduction, the authors appear to leave out the most well-studied size-sieving membranes for energy storage: microporous polymers. If we confine the discussion to flow batteries, then it would be appropriate to include citations to: Nature Materials 19, 195–202 (2020); Joule 3, 2968–2985 (2019); Adv. Energy Mater. 6, 1600517 (2016); Nano Lett. 15, 5724–5729 (2015). Some of these are also hydroxide ion conductors and have been used as selective/blocking membranes in aqueous Zn/Fe batteries, which is particularly relevant to the discussion of the state-of-the-art in this manuscript.

Response: First, we appreciate the reviewer’s valuable suggestions. As suggested, we have modified and enriched the Introduction of the manuscript in the revised version. And the citations were also added in the revised version (Line 36, Page 2).

(2) In highlighting the performance of devices incorporating their membranes, they make bold claims as to the current density supported (200 mA/cm^2) as world-leading and a breakthrough enabled by the membrane. However, they do not provide references for comparison to what the leading edge might have been prior to this work. The authors will need to provide well researched documentation for this claim by citing the previous state of the art.

Response: We appreciate the reviewer’s very constructive comments. As suggested, we have concluded and compared the recently reported zinc-based flow battery systems as shown in the Supplementary Table 1 in the revised version (Line 245, Page 14). Compared with other zinc-based flow battery systems, indeed, our work demonstrated the highest EE and cycling stability ever reported at current density of 200 mA cm^{-2} .

Supplementary Table 1. Battery performance of recently reported zinc-based flow battery systems.

Aqueous flow battery	Membrane	Current density (mA cm^{-2})	Performance EE(%)	Cycles
Zn/Br ₂ ¹	polyolefin porous (PP)	160	80	100
Zn/I ₂ ²	PP/Nafion	80	72.8	500

Alkaline Zn/I ₂ ³	Nafion 117	20	70	10
Zn/TEMPO ⁴	fumasep F-930-RFD	80	50	—
Zn/Fe ⁵	Nafion	80	61.5	100
Neutral Zn/Fe ⁶	Microporous	25	68	120
Neutral Zn/Fe ⁷	Porous PBI	80	78	100
Alkaline Zn/Fe ⁸	Nafion 212	80	76	20
Alkaline Zn/Fe ⁹	Nafion 212	80	80-85	100
(our previous work)				
Alkaline Zn/Fe ¹⁰	PES/SPEEK	160	80	100
(our previous work)				
Alkaline Zn/Fe ¹¹	PBI	160	82.68	150
(our previous work)				
This work	LDH-G	200	82.36	400

(3) The authors are asked to differentiate better for the reader the theoretical results presented from those already described for superionic transport of hydroxide in LDHs.

Response: We thank the reviewer's very valuable comments. Previous theoretical explorations worked on the superionic transport of hydroxide in LDHs mainly focused on a vacuum environment with several water molecules adsorbed using the density functional theory (DFT) approach, which neglects the possible influences of an aqueous environment and temperature [Mater. Horiz., 2019, 6, 2087]. And our simulation focused on the dynamic behaviors of hydroxide transport under more realistic operating conditions, which provided a direct observation of the transport mechanism of hydroxide inside LDHs layers (See the supplementary movie). To clarify this, the comparison of our theoretical results with those already described for superionic transport of hydroxide in LDHs has been added in the revised manuscript (Line 184-188, Page 11).

(4) The authors seem overly confident that interlayer transport of ions through the gallery is the main mechanism governing the transport behaviors of different ions in the membrane, e.g., in Figure 3e and elsewhere. They neglect contributions arising from the LDH-Nafion composite architecture, which results in a multiplicity of interfaces, arrangements of LDH particles in the

composite, etc. In this way, their interpretation of the transport behaviors is too premature. To resolve this, it is necessary to provide evidence that the mechanism proposed and explored is dominant. This could involve, e.g., investigations of the properties of membranes with variable volume fraction of the LDH particles in the composite. Here, the authors would need to link structural outcomes of the compositing process with variable LDH volume fraction and the ensuing transport properties (conductivity, blocking ability, etc.). For example, at low volume fraction of LDH, the membrane should behave like Nafion. At high volume fraction, we recover the behavior described. In the intermediate range, it should be possible to assess the relative contributions of interfacial transport vs. within-LDH transport, perhaps also evidencing the volume fraction of LDH leading to percolation of those domains within the composite membrane. Linking this to theory is also important. For example, Nafion is a fluoropolymer. It could be that the interface between it and LDH is poorly defined, resulting in gaps between the polymer and the particles; these gaps often contribute to the transport behaviors observed. Alternatively, depending on the surface chemistry of the LDH, the interface can densify, forcing ions to diffuse within the interlayers, but experiencing high resistance as you cross between the interlayer, the densified Nafion, and onto the next LDH. The authors are further advised to consider the microstructural features of hydrated Nafion in the composite on the membrane transport properties. Thus, while the theoretical and characterization methods performed thus far are the logical extension from what was done previously, they focus too narrowly on diffusivity within interlayer, which may only be a minor contributor to the overall transport mechanism in a composite with Nafion.

Response: We appreciate the reviewer's valuable comments. Firstly, we should explain that a less amount of 1 wt.% Nafion here was chosen as a binder, and the mass ratio of LDHs/Nafion is 8/2. To enhance the interfacial adherence between the top layer and the support layer, a binder is essential for enhancing the stability of the composite membrane. Using Nafion as the binder during a simple coating process for preparing composite membranes can be found in early research, such as *Angew. Chem. Int. Ed.* 2016, 55, 3058–3062; *Int. J. Hydrogen Energy*, 2017, 42, 21806-21816, etc. Actually, binders like Nafion, PVDF, PTFE resins were widely used in fuel cells and batteries to improve the bonding of particles to substrate (*Angew. Chem. Int. Ed.*, 2011, 50, 3520-3524; *Angew. Chem. Int. Ed.*, 2011, 50, 2999-3002; *Nature*, 2001, 414, 345-352; *Adv. Mater.* 2020, 32, 2004240, etc). And the ratio of 8/2 (particles/binders) is commonly used and considered to be an appropriate ratio.

Following the reviewer's suggestions, to further identify the effect of LDH on ions transport behavior of the membrane, we gradually increased the proportion of Nafion binder and regulated the ratio of LDH : Nafion to 4:4, 2:8, 0:1. First, the morphology of the membrane with different ratios of LDHs/Nafion were characterized as shown in Supplementary Fig. 9. As the proportion of Nafion binder increases (the content of

inorganic LDHs nanoparticles decreases), the cross-sectional morphology of the coating layer gradually becomes thinner. When only Nafion binder (without LDHs) was involved, the coating-layer was completely made up of Nafion polymer with a thickness of 200 nm.

Supplementary Figure 9. The cross-section morphology of prepared membranes with different ratio of LDHs/Nafion. a, LDH-M (4:4). b, LDH-M (2:8). c, LDH-Nafion (0:1).

To identify the hydroxide ion transport behaviors for different membranes, the hydroxide ion transference number through the membrane was analyzed and calculated by measuring the current-voltage (I-V) curve in a NaOH concentration gradient of 1|3 (Supplementary Fig. 10). The Na^+ and OH^- transference numbers calculated from Nernst equation for LDH-M (8:2) is 0.23 and 0.77, respectively, indicating that the OH^- acts as the main charging-balancing ions for the designed LDH-M. It is known that Nafion is a typical representative for cation exchange membrane, it mainly conducts cations through the anion exchange groups in alkaline media (J. Membr. Sci. 2018, 566, 8–14). LDHs can guarantee the fast OH^- transport through the hydrogen bond network among the hydroxyl groups, interlayer anions (OH^-) and water molecules in the gallery. As the proportion of Nafion increases (the content of LDHs decreases), the OH^- transference numbers gradually decreased from 0.77 to 0.66 for LDH-M (8:2), LDH-M (4:4) and LDH-M (2:8), respectively. As expected, the OH^- transference number for LDH-Nafion (0:1) reversed to be 0.37, which indicates that the main charging-balancing ions become Na^+ , the membrane behaves like Nafion. At the low proportion of Nafion, the prepared LDH-M follows the ion transport mechanism as LDHs does. While with the proportion of Nafion increased, the membranes gradually follow the cationic transport property as Nafion does. This proves that in the prepared LDH-M, hydroxide ions are mainly transferred through the channels of the LDHs (Line 128-134, Page 7). Low hydroxide ion transfer resistance is also a feature of LDHs, which can realize rapid OH^- transport through the hydrogen bond interactions in LDHs.

Supplementary Figure 10. The hydroxide ion transference numbers through different membranes (prepared from different ratio of LDHs/Nafion) calculated from the current-voltage (I-V) profiles.

Further, the corresponding battery tests using the aforementioned membranes were conducted to identify the high performance of LDH-M (Supplementary Fig. 17). Compared with the EE of the battery with a substrate, the battery assembled with the LDH-M exhibited a much higher EE of 89.1% (Line 228-229, Page 14). As the proportion of Nafion increase, the OH^- transference resistance increases, thus, the VE of the battery decreases and consequently, the EE of the battery decreases. The higher resistance can result in a higher ohmic polarization of the battery, which could afford the battery with a higher charge voltage and lower discharge voltage. The battery with the LDH-Nafion (with higher membrane resistance) will reach the cut-off voltage (0.1 V) earlier in the discharge process due to the high ohmic polarization, resulting in the battery with a lower VE compared to that of the battery with a LDH-M membrane (the charge capacity of the battery was kept constant).

Supplementary Figure 17. The AZIFB performances assembled with different membranes at the current density of 80 mA cm^{-2} .

Based on the above clarifications and identifications, we can conclude that the hydroxide ion transport through LDH-M is mainly realized along the interlayer of LDHs. Actually, we cannot neglect the existence of gaps and Nafion binder, and it is inevitable in applying nanomaterials to battery devices or fuel cells (Angew. Chem. Int. Ed., 2011, 50, 3520-3524; Angew. Chem. Int. Ed., 2011, 50, 2999-3002; Nature, 2001, 414, 345-352; Adv. Mater. 2020, 32, 2004240). Even though, the gaps and Nafion binder have little contribution to the ion transport of the membrane. (To clarify this, the corresponding discussions were added in the revised version, Line 128-134, Page 7)

(5) It would be useful to evaluate the [hydroxide]-dependent conductivity of the membrane. This also serves to distinguish the properties of the membrane for alkaline flow batteries relative to Nafion. Currently, the data presented in Figure 3 does not yet establish these membranes as fast ion conductors for hydroxide relative to the state of the art. Figure 3d, in particular, may just as well arise from a defective membrane (i.e., with an aberration or tear). To convince the reader, it is necessary to evaluate the conductivity of the membrane with variable area, thickness, etc. To convincingly demonstrate there is not a tear or other aberration, it is necessary to measure the Gurley number. To convincingly demonstrate the membrane can be handled without being damaged, the authors would need to evaluate the mechanical properties of the membrane, as is typical for Nafion and similar membranes for flow batteries and fuel cells.

Response: We greatly appreciate the reviewer for providing valuable suggestions to improve the quality of our manuscript. To elevate the accidental errors, three samples were tested for each kind of membrane. The error bars were shown in Fig. 3d in the

revised version. During the process of testing, to eliminate the contact resistance, electric resistance of the device and errors arisen from the defective membranes, several pieces of membranes were stacked together and a relationship between the resistance and the number of membranes stacked was constructed. One membrane sample was divided into 4 pieces and we measured the resistance of 1 to 4 pieces membranes stacked together and got the resistance of one membrane by calculating the slope of total resistance vs. the number of membranes stacked. An example is provided in Supplementary Fig. 7. The conductivity of LDH-M was on the order of 10^{-2} S cm^{-1} and approaching 10^{-1} S cm^{-1} , which is higher than that of the commercial membranes (on the order of 10^{-3} S cm^{-1}).

Figure 3d. Ionic conductivity of different membranes at 298 K with error bars.

Another important manifestation for fast hydroxide ion conductors of LDHs was the electrochemical performance of alkaline zinc-iron flow battery (AZIFB). The AZIFB assembled with LDH-M demonstrated high energy efficiency (EE, 82%) at the current density of 200 mA cm^{-2} , which was attributed to the fast hydroxide ion conductivity and the enhanced ion selectivity of LDHs, thus, bringing a worth-noting battery performance. Furthermore, we have conducted the ion transference number tests to identify the OH^- transport behavior as shown in Fig. 3e in the revised version. The OH^- transference number calculated from the current-voltage curve for LDH-M is 0.77, indicating that the OH^- acts as the main charging-balancing ions for the designed LDH-M. In comparison with the OH^- transference number of the LDH-M, the OH^- transference numbers of Nafion212, Nafion 115 and PBI were also calculated (Line 124-126, Page 7). As shown in Supplementary Fig. 8, the results show that the PBI membrane demonstrates almost the same anion and cation transference numbers. And the commercialized Nafion series cation exchange membranes tend to conduct cation (Na^+) in a NaOH concentration

gradient of 1/3. The OH^- transference numbers for Nafion 115 and Nafion 212 is 0.25 and 0.12, respectively. The results confirmed the high hydroxide ion conductivity of LDH-M compared with that of Nafion series membranes and PBI membrane.

Supplementary Figure 8. The hydroxide ion transference numbers through different membranes calculated from the current-voltage (I-V) profiles.

Gurley number is normally used in gas separation membrane to characterize gas permeability, it is note that LDH-M here was designed for ions sieving/transport. We are very sorry that currently we cannot directly find the gurley testing instrument. Instead, we detect the separation factor of LDH-M by using N_2/CO_2 as the feeding gas. The separation factor of LDH-M for N_2/CO_2 is 9, while the separation factor of the substrate is 1. Furthermore, we measured the hydrogen gas permeability through the LDH-M membrane with a gas chromatograph at 0.05 Mpa, the H_2 permeability through the LDH-M is $0.015 \text{ cm}^3/\text{cm}^2 \cdot \text{min}$. Compared with the substrate membrane, LDH-M demonstrates higher selectivity.

To convincingly demonstrate the membrane can be handled without being damaged, the mechanical stability of the membranes were measured. As seen in Supplementary Table 2 in the revised manuscript (Line 246-249, Page 14), the prepared LDH-M demonstrated a tensile strength of 12.36 MPa, which was comparable to that of the substrate and slightly lower than that of the Nafion series membranes. This is attributed to the dense property of Nafion series membranes, and porous membranes with high porosity normally exhibited relatively low mechanical performance. To identify the mechanical stability of LDH-M, the mechanical stability of LDH-M after more than 400 cycles battery test at the current density of 200 mA cm^{-2} (LDH-M-Cycle) was tested, the almost similar values of mechanical performance for LDH-M-Cycle indicated the good mechanical stability of LDH-M.

Supplementary Table 2. Mechanical performance of different membranes.

Membrane	Elongation at break (%)	Breaking stress (MPa)	Tensile strength (MPa)
Nafion 212	83.11	24.58	24.58
Nafion 115	119.28	20.36	20.36
Substrate	37.56	12.98	13.13
LDH-M	37.94	12.19	12.36
LDH-M-Cycle	33.72	12.12	12.16

Furthermore, XRD technology was performed on LDH-M-Cycle (Supplementary Fig. 22, Line 249-252, Page 14 in the revised version). As seen, the characteristic peaks of (003) and (006) still remain for the LDH-M after more than 400 cycles battery test (LDH-M-Cycle) at a current density of 200 mA cm^{-2} , which indicated the excellent anti-alkali stability and mechanical stability of LDH-M. Also, the SEM was carried out on the membranes after more than 400 cycles test at the current density of 200 mA cm^{-2} (Supplementary Fig. 22), the cross-section and surface morphologies identified the LDHs layer remained very stable, indicating the stability of the LDHs-coated layer on the substrate.

Supplementary Figure 22. XRD patterns of MgAl-CI-LDH, Substrate, LDH-M and LDH-M-Cycle.

(6) The Zn/Fe redox-flow cells have been demonstrated with low volumetric capacity ($\sim 15 \text{ Ah/L}$) yet comparable volumetric energy density ($\sim 25 \text{ Wh/L}$) to the more recognizable

vanadium redox-flow batteries. If the membrane is selective as purported, it should be possible to increase those significantly and elevate the potential for impact. Many emerging flow battery chemistries can go even higher on those fronts.

Response: Normally, the increased selectivity of the membrane brings higher coulombic efficiency (CE) for the battery. In theoretical calculation, CE is the ratio of discharge capacity to charge capacity during the same cycle. It reflects the reversibility of the battery. While, the volumetric capacity of a flow battery is determined by the electrolyte concentration and the number of electron transfer. And, the volumetric energy density of a flow battery is determined by the volumetric capacity and battery voltage. These are the intrinsic characteristics for battery systems. And, the volumetric energy density and volumetric capacity of a flow battery are not directly related to membrane's selectivity.

(7) The authors are respectfully asked to avoid using nebulous terms such as "Base" when describing any control membranes in the flow cell work. Just specify what "Base" actually means. In particular, the authors should provide to the reader as assessment of the flow cells where Nafion is used (i.e., as a negative control where the LDH is not featured in the membrane) and where a state-of-the-art membrane (e.g., PBI as a positive control) is used. It appears from their previous work that this has been done before and could provide the reader here with useful information needed to understand the progress made and enabled by the LDH component in the membrane.

Response: Sorry for the misunderstanding of using "Base", the "Base" was changed as the "Substrate" in the revised version.

As suggested, we conducted the battery tests using Nafion 212, Nafion 115 and PBI as the membranes for comparison (Supplementary Fig. 21, Line 243-244, Page 14 in the revised version). At a current density of 200 mA cm⁻², an AZIFB equipped with LDH-M demonstrated an energy efficiency (EE) of 82.4%, which was much higher than that of the battery with Nafion 212 (63.6%), Nafion 115 (60.6%), PBI (75.8%) and porous substrate (73.9%). Overall, the AZIFB assembled with LDH-M membranes exhibited significant high efficiencies than those of Nafion series membranes, PBI and substrate, which results from the combination of high ion selectivity and conductivity of LDH-M.

Supplementary Figure 21. Efficiencies of AZIFBs equipped with different membranes at the current density of 200 mA cm^{-2} .

(8) The authors are asked to consider and relate to the reader more directly what specific problem or application in grid storage these flow battery cells might be addressing.

Response: Electrical energy storage technologies attracts wide attention as the demands of efficient clean energy increasing, especially renewable energies. The basic function of electrical energy storage technologies is to store electricity, while still release it as needed. Combining electrical energy storage technology with renewables could effectively solve the most critical issue of their random and intermittent nature, which makes them quite challenging for their integration into the grid. The flow batteries are well suitable for large scale energy storage with their best combination of security, efficiency and flexibility. For energy storage systems, the most important factors are power density and energy density. Under normal circumstances, the energy density of flow battery depends on the characteristic of active materials, and the energy is proportional to the volume of the electrolyte used in energy storage tank. Compared with energy storage system of lithium-ion battery, the energy density of flow batteries is relatively low. This requires flow batteries to be able to operate at high current density to achieve high power density. Therefore, maintaining excellent battery performance under high current density is

particularly important. As suggested, this description was added in the introduction of revised manuscript (Line 28-30, Page 2).

Response to Reviewer 2:

Comments to the Author

This work reports a LDH membrane having a high OH^- conductivity and high ion selectivity and its use for an alkaline flow battery. The new insight that this work provides would be fast OH transport via Grotthous mechanism based on MD simulation, which is facilitated by the LDH structure. The performances of the membrane are interesting and meaningful in this technology sector, but the physics regarding the ion, and mass transport need to be further clarified.

Response: Firstly, we appreciate the reviewer's very positive feedbacks on our work

1) The structure of base membrane should be described in more detail. Why is PES selected as a base? Porosity and pore size distribution would be needed. Are the finger-like macropores the major channel for mass transport in the base membrane? If so, is there any problem of inhomogeneous flux? The interface between LDH and base membrane is strong enough? Is the interfacial stability evaluated?

Response: We greatly appreciate the reviewer's very valuable comments. As suggested, the structure of the membrane was further characterized by SEM (Supplementary Fig. 3). As seen in Supplementary Fig. 3a, the PES substrate shows an asymmetric morphology, which is composed of a dense skin layer and a typical finger-like macrovoids. Actually, PES asymmetric membranes were widely used for separation, where the finger like pores provide membranes with mechanical stability and very thin skin layer will ensure selectivity. In this paper, PES was selected as support, due to its super stability under alkaline medium. And the high porosity and large pores generally offer the membranes with very high conductivity. While, introducing a highly selective LDHs can improve the selectivity dramatically, as a result, to form a highly selectivity composite membranes.

Supplementary Figure 3. The cross-section morphology of PES substrate. a, The overall analysis image of substrate. The magnified morphology of the substrate marked at different locations. b, The upper pore morphology. c, Pore morphology in finger-like pores. d, The bottom pore morphology.

The porosity of the PES substrate measured by weighting method is 61.4%. Furthermore, BET test was conducted to identify the pore size distribution of the substrate (Supplementary Fig. 4). The results show the pore size distribution of the substrate is tens of nanometers, which is in accordance with the SEM results (Supplementary Fig. 3c). Indeed, the finger like pores here could offer major channels for mass transport, however, connected with nano pores on pore walls, therefore, there are no problems of inhomogeneous flux. (Line 91-93, Page 5 in the revised version).

Supplementary Figure 4a. The pore size distribution curve of the substrate.

For the interface issues, to strengthen the interface between LDH and substrate, a very minor amount of binder (Nafion solution, PTFE, PVDF), which was widely used in fuel cells, solar cells and the stability was well confirmed (Angew. Chem. Int. Ed. 2016, 55, 3058–3062; Int. J. Hydrogen Energy, 2017, 42, 21806-21816; Angew. Chem. Int. Ed., 2011, 50, 2999-3002; Nature, 2001, 414, 345-352; Adv. Mater. 2020, 32, 2004240, etc) To further confirm the interfacial stability, the morphologies of the membranes after 400 cycles test at a current density of 200 mA cm^{-2} (LDH-M-Cycle) was carried out. (Supplementary Fig. 22), the cross-section and surface morphologies identified the LDHs layer remained very stable on the substrate, indicating that the interface between LDH and substrate membrane is very stable. 200 mA cm^{-2} (LDH-M-Cycle)

Furthermore, XRD technology was carried out on LDH-M after cycling of the battery (Supplementary Fig. 22 (Response to reviewer 1)). As seen, the characteristic peaks of (003) and (006) still remain for the LDH-M after more than 400 cycles battery test (LDH-M-Cycle) at the current density of 200 mA cm^{-2} , which indicated the excellent anti-alkali stability and mechanical stability of LDH-M (Line 249-252, Page 14 in the revised version).

2) Although the authors state that the LDH particles are well aligned in the membrane (30micron), many interstitial pores appear. What is the porosity and pore size distribution of the LDH layer? Can the contribution from the pore be neglected in the conductivity analysis? Or is the membrane dense due to the Nafion binder? Since the nano channels of a single particle are not aligned with each other (It would be not possible), the ions should find their way from the edge of one particle to that of nearby particle. Therefore, the interstitial pore may have its effect on transport. In the case that interstitial pores are connected with a lower resistance,

molecules would pass through the pores instead of nanochannel. Compression (densification) of LDH membrane would show the effect of the interstitial pores.

Response: We really appreciate the reviewer for providing insightful comments on our work. Currently, we cannot accurately separate the thin LDHs layer from the substrate, so we tested the overall porosity of the LDH-M by weighting method. The porosity of LDH-M is 64.1%, which is relatively higher than that of the substrate (61.4%). The increased porosity resulted from the LDHs layer on the porous substrate, which brings more nanopores and higher porosity for LDH-M. BET test was conducted to identify the pore size distribution of LDH-M (Supplementary Fig. 4b). The results show the pore size distribution of the substrate is tens of nanometers (Supplementary Figure 4a). By contrast, a new distribution of micropores that corresponds to the nanochannels of LDHs appeared. And, benefitting from the presence of these micropores, LDH-M showed higher adsorption capacity in the microporous and mesoporous regions, thus, endowing the LDH-M with a higher specific surface area (Supplementary Fig. 4c, Line 96-97, Page 6 in the revised manuscript).

Supplementary Figure 4. (b) Pore size distribution curve of LDH-M. (c) N_2 isothermal absorption/desorption curves of LDH-M and substrate.

Actually the LDHs were stacked together by binders, which will lead to a defect free membranes, which can be confirmed very well by very impressive battery performance. Indeed, the effect of binders at the edge of LDH on the transport can not be clarified very clearly. Nafion binder plays an important role in linking LDH nanoparticles on the substrate. However, the transmission resistance of ions through Nafion is higher than that in the LDHs interlayers, since Nafion normally transport cations through under alkaline medium, which results in the alkaline zinc-iron flow battery with a bit lower voltage efficiency. By contrast, the hydrogen bond network in LDH nanochannel has been proven to be able to realize fast hydroxide ions transport. Although there are some interstitial pores on the surface of LDH-M, the main ion transport is achieved through LDHs. To identify the hydroxide ion transport behaviors of LDH-M, we fabricated the

membranes with different ratios of LDH / Nafion (4:4, 2:8, 0:1). The hydroxide ions transference number through the membrane was analyzed and calculated by measuring the current-voltage (I-V) curve in a NaOH concentration gradient of 1/3 (Supplementary Fig. 10 (Response to reviewer 1)). The Na^+ and OH^- transference numbers were calculated from Nernst equation for LDH-M (8 : 2) is 0.23 and 0.77, respectively, indicating that the OH^- acts as the main charging-balancing ions for the designed LDH-M. As the proportion of Nafion binder increases (the content of LDHs decreases), the OH^- transference numbers gradually decreased from 0.77 to 0.66 for LDH-M (8:2), LDH-M (4:4) and LDH-M (2:8), respectively. Particularly, the OH^- transference number for LDH-Nafion (0:1) reversed to be 0.37, which indicates that the main charging-balancing ion becomes Na^+ , and the membrane behaves like Nafion. At the low proportion of Nafion, the prepared LDH-M follows the ion transport mechanism as LDHs does. This proves that in the prepared LDH-M, the hydroxide ions are mainly transferred through the channels of the LDHs. To clarify this, more discussion was added in the revised version (Line 128-134, Page 7 in the revised manuscript).

Further, the corresponding battery tests using the aforementioned membranes were conducted to identify the high performance of LDH-M (Supplementary Fig. 17(Response to reviewer 1)). Compared with the EE of the battery with a substrate, the battery assembled with the LDH-M exhibited a much higher EE (89.1%). As the proportion of Nafion increase, the OH^- transference resistance increases, the VE of the battery decreases and the EE of the battery decreases accordingly. The higher resistance can result in a higher ohmic polarization of the battery, which could afford the battery with a higher charge voltage and lower discharge voltage. The battery with the LDH-Nafion (with higher membrane resistance) will reach the cut-off voltage (0.1 V) earlier in the discharge process due to the high ohmic polarization, resulting in the battery with a lower efficiency compared to that of the battery with a LDH-M membrane (Line 228-229, Page 13 in the revised manuscript).

Based on the above clarifications and identifications, we can conclude that the hydroxide ion transport through LDH-M is mainly realized along the interlayer of LDHs. Indeed, we cannot neglect the influence of the interstitial pores (formed by the stacked LDHs) and Nafion binder on the hydroxide ion transport, and it is also an inevitable issue in applying nanomaterials to battery devices or fuel cells (Angew. Chem. Int. Ed., 2011, 50, 3520-3524; Angew. Chem. Int. Ed., 2011, 50, 2999-3002; Nature, 2001, 414, 345-352; Adv. Mater. 2020, 32, 2004240). Even though, the interstitial pores and Nafion binder are proved to have little contribution to the ion transport of the membrane. (To clarify this, the corresponding discussion was added in the revised version, Line 128-134, Page 7)

3) Is the selectivity of LDH membrane higher than Nafion 212, PBI, 115 membrane?

Compare the transference number for LDH, Nafion and PBI also.

Response: We thank the reviewer's very constructive comments. As suggested, the selective ions transport behaviors of different membranes were verified by using concentration-driven dialysis diffusion tests (Fig. R1). The results show that the order of the ionic transport rate through LDH-M, substrate and PBI is $K^+ > Na^+ > Ca^{2+} > Mg^{2+} > Fe(CN)_6^{3-}$. In particular, it is known that Nafion and PBI are the typical representative of ion exchange membranes, they mainly conduct ions through the ion exchange groups in an ion exchange mechanism rather than selective screening in a pore size exclusion mechanism. Combined with the dense property and ion exchange mechanism of Nafion series membranes and PBI membrane, they demonstrate higher ion selectivity for ions than those of porous membranes. Nevertheless, the LDH-M demonstrated higher selectivity than does the substrate.

Figure. R1 Selective ions permeation of common salts through different membranes.

It is note that as for $Fe(CN)_6^{3-}$, the LDH-M shows the same order of ionic transport rates as that of dense membrane like Nafion series and PBI membranes (Fig. R2). A significant lower $Fe(CN)_6^{3-}$ ionic transport rate (higher ions selectivity) than that of the substrate can be achieved (Fig. R12). This means the LDHs selective layer does play a decisive role in membrane's ion selectivity, especially for $Fe(CN)_6^{3-}$. Thus the LDHs can efficiently impede active species ($Fe(CN)_6^{3-}$) while transfer the charge carrier (OH^-) for AZIFBs.

Figure. R2 The permeability of $\text{Fe}(\text{CN})_6^{3-}$ through different membranes.

Furthermore, the ion transference number of PBI and Nafion series membranes in a NaOH concentration gradient of 1|3 was also measured and calculated (Supplementary Fig. 8 (Response to reviewer 1)). The OH^- transference number calculated from the I-V curves for LDH-M is 0.77, indicating that the OH^- acts as the main charging-balancing ions for the designed LDH-M. Meanwhile, the PBI ion exchange membrane demonstrated almost the same anion (OH^-) and cation (Na^+) transference number. Nevertheless, the OH^- transference number for Nafion 115 and Nafion 212 is 0.25 and 0.12, respectively, indicating that the commercialized Nafion series cation exchange membranes mainly conduct cation (Na^+) in alkaline media. The above results confirmed the high hydroxide ion conductivity of LDH-M in comparison with that of PBI and Nafion series membranes (Line 124-126, Page 7 in the revised manuscript).

4) For MD simulation, how come the numbers of water and OH^- molecule are defined. Is it based on experiment or assumption? Is electric field applied to the set?

The existence of Grotthous mechanism is feasible. However, it is not clear how large its contribution in comparison with vehicular mechanism. Like nanocapillary action of water flow in graphene nanosheet (Science, 2014, 343, 752-754), capillary flow can happen.

Response: Thanks for the reviewer's insightful comments. For better simulating the transport behaviors of ions, the parameters was set based on the experimental conditions. The number of water molecules was calculated according to water density at atmospheric pressure (1 g cm^{-3}) by multiplying the free volume (enclosed by the Connolly surface with the Connolly radius set as 1.0 \AA). In order to keep the neutrality of charge of the simulated LDH system, the number of OH^- was determined based on the number of Mg^{2+} replaced by Al^{3+} . To observe the spontaneous transport behavior of hydroxide, the electric field was not considered during the *ab initio* molecular dynamics (AIMD)

simulation. The detailed and additional descriptions for MD simulation were added and provided in the revised version (Line 376-379, Page 21).

As for the contribution of Grotthuss and vehicular mechanism, we calculated the MSD of OH^- which accounts for the movement induced by both the Grotthuss and vehicular mechanisms (Nat. Commun. 2020, 11, 13), as well as the MSD of O atoms (in water and hydroxide ions) which only qualitatively represents contribution of vehicular mechanism. As depicted in Supplementary Fig. 16, the MSD of O atoms (vehicular mechanism) was much smaller than that of OH^- (Grotthuss and vehicular mechanisms), demonstrating that the fast transport of OH^- in the LDHs layers was mainly contributed by the Grotthuss mechanism.

Supplementary Figure 16. Mean square displacement (MSD) of hydroxide ions and all O atoms in water and hydroxide ions in LDHs at 298 K.

5) In comparison of base and LDH-base membrane, base cell showed slightly higher voltage efficiency. Why? the ohmic resistance of membrane should be larger for LDH-base membrane because LDH layer is added to the base. In comparison with Nafion or PBI membrane, does LDH membrane have higher coulombic and energy efficiency?

Response: We appreciate the reviewer's attention to the details. Indeed, the ohmic resistance of membrane is larger for LDH-M membrane because an inorganic LDH layer is introduced to the substrate (Fig. 3d). The AZIFB assembled with LDH-M delivered an enhanced coulombic efficiency (CE) and slightly lower voltage efficiency (VE) compared to that of the battery with substrate. Thus, the energy efficiency (EE) of AZIFB assembled with LDH-M has a significant improvement (Fig. 6a in the revised version). This improvement is attributed to the coating layer of LDHs that can enhance the ion

selectivity of the membrane and keep almost similar high hydroxide ion conductivity (Fig. 3c in the revised version).

As we response to Reviewer 1, we conducted the battery tests using Nafion 212, Nafion 115 and PBI as the membranes for comparison. As shown in Supplementary Fig. 21 (Response to reviewer 1), at a current density of 200 mA cm^{-2} , an AZIFB equipped with LDH-M demonstrated an energy efficiency (EE) of 82.4%, which was much higher than that of the battery with PBI, Nafion 212 (EE), Nafion 115 (EE) and porous substrate (EE). Overall, the AZIFB assembled with LDH-M exhibited much higher efficiencies over Nafion series membranes, PBI and substrate, which results from the combination of high ion selectivity and conductivity of LDH-M (Line 243-244, Page 14 in the revised manuscript).

Response to Reviewer 3:

Comments to the Author

This work reported a novel layered double hydroxides (LDHs) composite membrane with a high selectivity and superb hydroxide ion conductivity for alkaline-based flow batteries. The transport behavior of OH^- in restricted interlayer gallery of LDHs was clearly presented and clarified. The contribution created the application of LDHs nanomaterials in alkaline-based flow batteries and achieved very encouraging performance ever reported. This is an impressive work in the field of size-based separation materials and alkaline based batteries. The work will provide very important information on understanding of transport mechanism of OH^- in LDHs nanomaterials, as well as on both 2D materials and energy storage devices. Therefore, I recommend accepting this paper in Nature Communications after considering the following minor comments:

Response: **We appreciate the reviewer's very positive feedbacks on our work**

1. The anti-alkaline stability of layered double hydroxide membrane (LDH-M) under the complex environment of electric field and strong alkali after battery cycling should be verified by XRD results though the stability of LDHs in alkaline environment was presented (Figure 2g).

Response: **We appreciate the reviewer's very valuable suggestions. As suggested, we conducted the XRD tests after battery cycling to demonstrate the anti-alkaline stability of LDH-M (Supplementary Fig. 22 (Response to reviewer 1)). The results show that the characteristic peaks of LDH remains for the LDH-M after more than 400 cycles test (LDH-M-Cycle) at a current density of 200 mA cm^{-2} , which indicated the excellent anti-alkaline stability of the prepared LDHs. Furthermore, the SEM was carried out on the membrane after more than 400 cycles test at the current density of 200 mA cm^{-2} . The cross-section and surface morphologies identified the LDHs layer remained very stable,**

indicating the stability of the LDHs coating layer on the substrate as well (Line 246-252, Page 14 in the revised manuscript).

2. The charge-discharge profiles at the end of charging for Base appears a sudden arise in Figure 6b, the authors need to comment on this.

Response: The sudden arise of charge profiles corresponded to the concentration polarization of the battery assembled with Base (substrate), which can also be observed in Fig. 6e at the end of charging of the AZIFB in the revised version. Because of the lower selectivity of Base, the electrolyte tends to crossover during charging-discharging of the battery, which lowers the concentration of the active materials for both positive and negative electrolytes. The lowered concentration of active materials (Zn(OH)_4^{2-} and $\text{Fe(CN)}_6^{4-}/\text{Fe(CN)}_6^{3-}$) in both sides of the electrolyte further resulted in a high concentration polarization of the battery. As identified in Fig. S20 in the revised version, the color of negative electrolyte changed to be yellow, resulting from the cross-over of positive electrolyte ($\text{Fe(CN)}_6^{4-}/\text{Fe(CN)}_6^{3-}$).

3. From the area resistance results (Figure 3d), LDH-M seem to present a higher area resistance, which should lead to lower voltage efficiencies, but higher VEs and EEs are achieved in the battery tests (Figure 6a). Pls clarify this?

Response: We appreciate the reviewer's attention to the details. The ohmic resistance of LDH-M is larger than does the Base since an inorganic LDH layer was introduced to the Base (Fig. 3d), which enables a longer ions transport path. The AZIFB with LDH-M delivered an enhanced coulombic efficiency (CE) and similar voltage efficiency (VE) compared to that of the battery with substrate. As a result, the energy efficiency (EE) of AZIFB assembled with LDH-M has a significant improvement (Fig. 6a in the revised version). This improvement is attributed to the coating layer of LDHs on the substrate that can enhance the ion selectivity of the membrane and keep almost similar high hydroxide ion conductivity (Fig. 3c in the revised version).

4. For the DFT calculation part, the function and basis set should be provided, and the selectivity of Fe(CN)_6^{3-} for DFT calculation as mentioned should be included in the SI. Also, please explain the reason why including model-2 and model-3 in this simulation.

Response: We thank for the reviewer's valuable suggestions. The initial structure was optimizing by DFT calculation, the function and the basis set were the same as that used in AIMD, which have been provided in the revised manuscript (Line 384-385, Page 21).

As the reviewer's suggestion, the selectivity of Fe(CN)_6^{3-} was also added in the revised manuscript by comparing the MSD values with that of OH^- in LDHs. As can be seen in Fig. 3f in the revised version, similar to the case of Fe(CN)_6^{4-} , Fe(CN)_6^{3-} was also difficult to diffuse into interlayer gallery of LDHs as evidenced by its low MSD values.

Figure 3f. Mean square displacement (MSD) of hydroxide, zincate, ferrocyanide, ferricyanide ions in LDHs at 298 K.

The model-2 and model-3 with different ions (zincate and ferricyanide ions, respectively) were optimized and AIMD simulations were further performed to calculate the mean square displacement (MSD) as illustrated in Fig.3f in the revised version, in order to compare the selectivity of LDHs to OH^- than $\text{Zn}(\text{OH})_4^{2-}$ and $\text{Fe}(\text{CN})_6^{4-}$.

5. The experimental details of assembling the flow battery should be provided.

Response: As suggested, more experimental details were added in the revised version (Line 360-361, 364, Page 20).

6. The format through the manuscript should be carefully checked and improved, e.g., the Ref. No oscillated before and after punctuations.

Response: Sorry for the confusion, as suggested, we have carefully checked and modified the format of our manuscript in the revised version.

Reference

- 1 Wang, C. *et al.* A TiN nanorod array 3D hierarchical composite electrode for ultrahigh-power-density bromine-based flow batteries. *Adv. Mater.*, e1904690, doi:10.1002/adma.201904690 (2019).
- 2 Xie, C., Liu, Y., Lu, W., Zhang, H. & Li, X. Highly stable zinc–iodine single flow batteries with super high energy density for stationary energy storage. *Energy Environ. Sci.* **12**, 1834-1839, doi:10.1039/c8ee02825g (2019).
- 3 Zhang, J. *et al.* An all-aqueous redox flow battery with unprecedented energy density. *Energy Environ. Sci.* **11**, 2010-2015, doi:10.1039/c8ee00686e (2018).

- 4 Winsberg, J. *et al.* Aqueous 2,2,6,6-Tetramethylpiperidine-N-oxyl catholytes for a high-capacity and high current density oxygen-insensitive hybrid-flow battery. *Acs Energy Lett* **2**, 411-416, doi:10.1021/acsenergylett.6b00655 (2017).
- 5 Gong, K. *et al.* A zinc-iron redox-flow battery under \$100 per kW h of system capital cost. *Energy Environ. Sci.* **8**, 2941-2945, doi:10.1039/c5ee02315g (2015).
- 6 S. Selverston, z. R. F. S. a. J. S. W. Zinc-iron flow batteries with common electrolyte. *J. Electron. Mater.* **164**, A1069-A1075, doi:10.1149/2.0591706jes] (2017).
- 7 Xie, C., Duan, Y., Xu, W., Zhang, H. & Li, X. A low-cost neutral zinc-iron flow battery with high energy density for stationary energy storage. *Angew. Chem. Int. Ed.*, doi:10.1002/anie.201708664 (2017).
- 8 MCBREEN, J. Rechargeable zinc batteries. *J. Electroanal. Chem.* **168** (1984).
- 9 Hu, J., Zhang, H., Xu, W., Yuan, Z. & Li, X. Mechanism and transfer behavior of ions in Nafion membranes under alkaline media. *J. Membr. Sci.* **566**, 8-14, doi:10.1016/j.memsci.2018.08.057 (2018).
- 10 Yuan, Z. *et al.* Negatively charged nanoporous membrane for a dendrite-free alkaline zinc-based flow battery with long cycle life. *Nat. Commun.* **9**, 3731, doi:10.1038/s41467-018-06209-x (2018).
- 11 Yuan, Z., Duan, Y., Liu, T., Zhang, H. & Li, X. Toward a low-cost alkaline zinc-iron flow battery with a polybenzimidazole custom membrane for stationary energy storage. *iScience* **3**, 40-49, doi:10.1016/j.isci.2018.04.006 (2018).

REVIEWER COMMENTS

Reviewer #1 (Remarks to the Author):

The authors use imprecise language in one of their central claims: "As a platform for verifying the practicability of LDHs-based membrane, an alkaline zinc-iron flow battery (AZIFB) assembled with the designed membrane was investigated in detail, demonstrating a high coulombic efficiency (CE) of over 98% and an energy efficiency (EE) of over 82% at a current density of 200 mA cm⁻², which is by far the highest value ever reported." Specifically, there are three values of importance: the CE, the EE, and the current density. It is unclear whether "The highest value ever reported" claim by the authors refers to any one of those metrics, some of them, or all of them. Please revise the text to clarify. In general, it is often recommended to authors not to assume that their efficiency, current density, etc. is a world record, best ever reported, first example of, etc. The authors should stand on the science of their work and not a presumed artifact of an imaginary race. The authors can report with distinction their results without resorting to inappropriate use of such superlatives.

The authors process the LDH layer by using spray coating of dispersed particles. Heterogeneity in thickness can be due to the coating process as well as the impact of the dispersant on the aggregated state of the particles, which can also change on drying. To understand this, the authors have cross-sectional SEM in Figure 2E, which shows heterogeneity of 75-100% in thickness of the selective transport layer on the substrate; note here, the EDX is not informative of thickness as much as panel B is, due to known issues with the EDX detector and depth perception. Consequently, their transport data (e.g., conductivity) accounting for heterogeneity in film thickness should have significant, non-negligible error noted in the associated data. This does not appear to be represented in the error in Figure 3. Importantly to the central claims of membrane selectivity for specific ions and for the conductivity of hydroxide ions: there are no error bars in the ion permeation data in Figure 3B; the error bars shown in Figure 3D for the conductivity data are likely grossly underestimated. Given the large error in LDH membrane thickness, these errors are large. Please revise to take into account the error associated with variable thickness of the LDH layers.

The term "ionic transport rate" is imprecise. Please use standard terminology to describe transport.

Perhaps I missed something important: why is the ion permeation rate of Zincate missing from Figure 3B? It's central to the Zn-Fe flow battery tested.

"with a gradient of 1|3 mol L⁻¹ NaOH solution" should show a range of numbers, e.g., 1-3.

Reviewer #2 (Remarks to the Author):

The revised version of the manuscript provides more detailed analysis on the effect of Nafion binder content, which more clearly demonstrate the transport process in the LDH membrane. The change of the H⁺ and OH⁻ transference number with Nafion content actually implies that not only LDH phase but also surrounding phase (Nafion + liquid electrolyte) contribute to the conduction. If the LDH is the only conducting phase, OH⁻ transference number will not be changed much while OH⁻ ionic conductivity is diminished. However, the excellent ion selectivity and cell performance implies that the ion transport through the surrounding phase (outside LDH) does not significantly contribute to the total ionic transport. In overall, the authors responses are reasonable and acceptable, thus, I support the publication of this work in Nature Communications.

Reviewer #3 (Remarks to the Author):

The author has carefully revised the manuscript and the reviewer recommend to accept this manuscript.

Response to Reviewer 1:

(1) The authors use imprecise language in one of their central claims: "As a platform for verifying the practicability of LDHs-based membrane, an alkaline zinc-iron flow battery (AZIFB) assembled with the designed membrane was investigated in detail, demonstrating a high coulombic efficiency (CE) of over 98% and an energy efficiency (EE) of over 82% at a current density of 200 mA cm⁻², which is by far the highest value ever reported." Specifically, there are three values of importance: the CE, the EE, and the current density. It is unclear whether "The highest value ever reported" claim by the authors refers to any one of those metrics, some of them, or all of them. Please revise the text to clarify. In general, it is often recommended to authors not to assume that their efficiency, current density, etc. is a world record, best ever reported, first example of, etc. The authors should stand on the science of their work and not a presumed artifact of an imaginary race. The authors can report with distinction their results without resorting to inappropriate use of such superlatives.

Response: First, we appreciate the reviewer's efforts on our work and thank the reviewer for providing insightful comments to further improve the quality of our manuscript. As shown in the Supplementary Table 1 in the revised version, compared with other zinc-based flow battery systems, indeed, our work demonstrated very high EE and cycling stability at the current density of 200 mA cm⁻². Generally, as the current density increases, the EE of the battery decreases. We compared the EE of the battery to demonstrate that this work can achieve high EE at the current density of 200 mA cm⁻². As suggested by the reviewer, the description was modified as "As a platform for verifying the practicability of LDHs-based membrane, an alkaline zinc-iron flow battery (AZIFB) assembled with the designed membrane was investigated in detail, demonstrating a high coulombic efficiency (CE) of over 98% and an energy efficiency (EE) of over 82% at a current density of 200 mA cm⁻², the achieved EE value is very high among recently reported zinc-based flow batteries" in the revised version.

(2) The authors process the LDH layer by using spray coating of dispersed particles. Heterogeneity in thickness can be due to the coating process as well as the impact of the dispersant on the aggregated state of the particles, which can also change on drying. To understand this, the authors have cross-sectional SEM in Figure 2E, which shows heterogeneity of 75-100% in thickness of the selective transport layer on the substrate; note here, the EDX is not informative of thickness as much as panel B is, due to known issues with the EDX detector and depth perception. Consequently, their transport data (e.g., conductivity) accounting for heterogeneity in film thickness should have significant, non-negligible error noted in the associated data. This does not appear to be represented in the error in Figure 3. Importantly to the central claims of membrane selectivity for specific ions and for the conductivity of hydroxide ions: there are no error bars in the ion permeation data in Figure 3B; the error bars

shown in Figure 3D for the conductivity data are likely grossly underestimated. Given the large error in LDH membrane thickness, these errors are large. Please revise to take into account the error associated with variable thickness of the LDH layers.

Response: Thanks for the comments. Indeed, the EDS mapping collected in corresponding SEM image (Figure 2e) is the same area of the membrane. Thus, the thickness is the same in theoretical. The thickness of the LDH on the substrate seems to be different from the SEM image (Figure 2e) and EDS mapping because the magnification and corresponding scale bar of the Figures are different (10 μm for SEM image and 60 μm for EDS mapping, respectively). The calculated thickness of LDHs is $15 \pm 1 \mu\text{m}$ for both two figures. To avoid confusion, we added the raw mapping figure in the revised manuscript (Figure 2f). During the EDS mapping test, the magnification of the figure was slightly zoom out to protect the membrane from damaging under high accelerating voltage shock, thus causing the different thickness in visual (Figure R1). Calculated from the scale bar, we can confirm that the thicknesses of the LDH layer on the membrane in Figure 2e and in EDS mapping are the same.

Figure 2. Layered double hydroxide composite membranes.

Figure R1. The raw datas in Figure 2e and the mapping.

Furthermore, we provided the repeated data of the cross-section morphology of the membrane in Figure R2, where the raw data and the repeated data are the cross-sectional morphology of two pieces of LDH-M membranes. They show the similar thickness of LDHs layer. This proves that the thickness of LDHs layer is universal and repeatable. During the preparation, we strictly control the amount, proportion and operation method of spraying to ensure the uniformity and repeatability of the membrane. As shown in Figure S5 in the revised version, the surface morphology is dense and uniform. Before the SEM tests, in order to obtain the membrane cross-section samples, the membrane was brittle fracture under liquid nitrogen. Under this process, it is easy to cause visual defects and unevenness under the cross-sectional morphology of the membrane, especially for the surface containing inorganic nanomaterials. Overall, the thickness of the LDHs is relatively uniform and repeatable.

Figure R2. The raw data in Figure 2e and the repeated data for the cross-sectional morphology of LDH-M.

Consequently, all the basic tests of the membrane shown in Figure 3 are repeatable and we have carried out repeated experiments and added the repeated data. As suggested, we

added the error bars of the data. Under the scientific notation axis of Figure 3b, the error bar is not obvious, which suggested the repeatability of the membranes.

Figure 3b. Ionic selectivity and conductivity of LDH-M.

There are many data points for the OH⁻ conductivity test, which made it difficult to make the error bars. Here, we provided the repeated data for the permeability of OH⁻, as shown in Figure R3.

Figure R3. (a) The raw data in Figure 3c and (b) the repeated data for the permeability of hydroxide ions.

The repeated data for Figure 3e was shown in Figure R4.

Figure R4. (a) The raw data in Figure 3e and (b) the repeated data for the ion transfer number of hydroxide ions.

Another important evidence reflecting the repeatability of the membrane is battery performance. If the membrane itself has unreasonable errors, there will be major difference in battery performances, and there will be no repeatability. Therefore, we added the error bars for battery performances, as seen, the battery performance for the membranes is repeatable.

Figure 6a. Rate performance as the current density ranging from 80 to 200 mA cm^{-2} . Consequently, these all datas comprehensively proved that the thickness of LDH layer on the prepared membrane is homogeneous and has no great influence on performance.

(3) The term "ionic transport rate" is imprecise. Please use standard terminology to describe transport.

Response: As suggested, we have corrected the description as the "ion permeation rate" in the revised manuscript (Line 109, 110, Page 6)

(4) Perhaps I missed something important: why is the ion permeation rate of Zincate missing from Figure 3B? It's central to the Zn-Fe flow battery tested.

Response: We appreciate the reviewer's valuable suggestion. As suggested, we conducted the ion permeation of zincate for LDH-M and substrate, the results was shown in Figure S7, where the LDH-M demonstrates much higher selectivity for zincate ion than that of the substrate as well (Line 112, Page 6).

Figure S7. The permeability of zincate ion for LDH-M and substrate.

(5) "with a gradient of $1/3 \text{ mol L}^{-1}$ NaOH solution" should show a range of numbers, e.g., 1–3.

Response: As suggested, we have corrected the description in the revised manuscript (Lines 147-148, Page 8).

Response to Reviewer 2:

The revised version of the manuscript provides more detailed analysis on the effect of Nafion binder content, which more clearly demonstrate the transport process in the LDH membrane. The change of the H^+ and OH^- transference number with Nafion content actually implies that not only LDH phase but also surrounding phase (Nafion + liquid electrolyte) contribute to the conduction. If the LDH is the only conducting phase, OH^- transference number will not be

changed much while OH⁻ ionic conductivity is diminished. However, the excellent ion selectivity and cell performance implies that the ion transport through the surrounding phase (outside LDH) does not significantly contribute to the total ionic transport. In overall, the authors responses are reasonable and acceptable, thus, I support the publication of this work in Nature Communications.

Response: We really appreciate the reviewer's efforts on our work and thank the reviewer for the positive feedbacks.

Response to Reviewer 3:

The author has carefully revised the manuscript and the reviewer recommend to accept this manuscript.

Response: We appreciate the reviewer's very positive feedbacks on our work.